# Theoretical relation between axon initial segment geometry and excitability

**Sarah Goethals, Romain Brette\***

Sorbonne Université, INSERM, CNRS, Institut de la Vision, Paris, France

**Abstract** In most vertebrate neurons, action potentials are triggered at the distal end of the axon initial segment (AIS). Both position and length of the AIS vary across and within neuron types, with activity, development and pathology. What is the impact of AIS geometry on excitability? Direct empirical assessment has proven difficult because of the many potential confounding factors. Here, we carried a principled theoretical analysis to answer this question. We provide a simple formula relating AIS geometry and sodium conductance density to the somatic voltage threshold. A distal shift of the AIS normally produces a (modest) increase in excitability, but we explain how this pattern can reverse if a hyperpolarizing current is present at the AIS, due to resistive coupling with the soma. This work provides a theoretical tool to assess the significance of structural AIS plasticity for electrical function.

## Introduction

Historically, the theory of neural excitability was developed on the space-clamped giant squid axon (*Hodgkin and Huxley, 1952a*), which stems from a syncytium (many cells merging their axons into a giant one) (*Young, 1936*). Spike initiation occurs in a different configuration in vertebrate neurons: spikes generally initiate in the axon initial segment (AIS), a small structure next to the soma (*Bender and Trussell, 2012*). Although this fact has been known for a long time (*Coombs et al., 1957*), it has been generally neglected in theory (*Koch, 1999*; *Tuckwell, 1988a*). Recently, its potential functional significance has been emphasized by a number of studies showing that both the position and length of the AIS can vary not only across cells (*Hamada et al., 2016*; *Höfflin et al., 2017*; *Kuba et al., 2006*), but also with activity (*Grubb et al., 2011*; *Jamann et al., 2018*). These observations raise the possibility that AIS movements could be involved in the regulation of excitability.

Variations of AIS geometry indeed co-occur with changes in excitability, but also with many other factors (*Kole and Brette, 2018*), such as changes in input resistance (*Grubb and Burrone, 2010*; *Hatch et al., 2017*; *Lezmy et al., 2017*; *Wefelmeyer et al., 2015*), phosphorylation of voltage-gated sodium (Nav) channels (*Evans et al., 2015*), redistribution of voltage-gated potassium (Kv) channels (*Kuba et al., 2015*), or changes in cell capacitance (*Kuba et al., 2014*). In some studies, distal displacement of the AIS is associated with decreased excitability (*Grubb and Burrone, 2010*; *Hatch et al., 2017*; *Lezmy et al., 2017*; *Wefelmeyer et al., 2015*). In others, neurons with more distal AIS have an identical (*Thome et al., 2014*) or slightly hyperpolarized threshold (*Hamada et al., 2016*). Thus, it is challenging to experimentally isolate the specific contribution of AIS geometry to excitability changes.

Modeling studies have also produced mixed results. Theoretical work proposed that increased electrical isolation of the AIS from the large soma should result in increased excitability (*Baranauskas et al., 2013*; *Brette, 2013*; *Telenczuk et al., 2017*). However, several numerical studies have reported decreased excitability when the AIS is moved distally, depending on cell morphology (*Gulledge and Bravo, 2016*) or on the expression of Kv channels in the AIS (*Lezmy et al., 2017*). These findings indicate that the relation between AIS position and excitability is highly nonlinear, since even the direction of change depends on model parameters.

**\*For correspondence:**
romain.brette@inserm.fr

**Competing interests:** The authors declare that no competing interests exist.

Finally, differences in excitability have been reported using various measures: rheobase, current density threshold or voltage threshold at the soma. It has also been argued that a more relevant quantity is the axonal threshold, since spikes initiate in the AIS (*Yu et al., 2008*). Thus, it is unclear how excitability should be characterized in order to capture the contribution of AIS geometry.

Here, we develop a principled theoretical analysis to address these issues and disentangle the different factors relating AIS geometry and excitability. In the first part, we examine the relation between cell geometry and passive properties of the soma-AIS system. When the axon is small compared to the somatodendritic compartment, an axonal current produces a larger local depolarization when it is applied further from the soma, while the converse holds when the axon is large. We show with neuroanatomical and electrophysiological data that the physiological situation is the former one. In the second part, we show that excitability changes caused by changes in the AIS are captured by the somatic voltage threshold (and not, perhaps counter-intuitively, by the AIS threshold). In the third part, we theoretically analyze the excitability of a spatially extended AIS with sodium channels, and in the fourth part, we consider the impact of non-sodium channels of the AIS (for example Kv7 channels). From this analysis, we derive a parsimonious mathematical expression for threshold variations as a function of AIS geometry, Nav conductance density and non-sodium current at the AIS. We find that, when the AIS is moved away from the soma, the cell becomes slightly more excitable, unless a strong hyperpolarizing current is present at the AIS. In the fourth part, we discuss the role of axon morphology, and finally in the fifth part we examine the relation with experimental observations.

## Results

### Passive properties

At spike initiation, a $Na^+$ current first enters the AIS, producing a local depolarization. How does this depolarization vary with the position of the injection site? As several studies have pointed out, the answer depends on the relative sizes of the soma (or somatodendritic compartment) and axon (*Brette, 2013*; *Eyal et al., 2014*; *Michalikova et al., 2017*; *Telenczuk et al., 2017*). In the following, we consider a current passing through the membrane of a passive cylindrical axon attached to the cell body, and we analyze two extreme cases: a very small soma, and a very large soma. We then show that at the time scale of spike initiation, the latter case approximates the physiological situation in many neuron types.

#### Small soma, or sealed end condition

We start with the theoretical case of a soma and axon of the same size, meaning that the neuron is simply a cylinder. In cable theory, this is called the 'sealed end condition': one end of the axon is sealed and no current passes through it (*Tuckwell, 1988b*). A current is injected at a distance $x$ from the soma, in an axon of space constant $\lambda$, typically about 500 µm in cortical pyramidal cells (*Kole et al., 2007*). The ratio between local depolarization and current is by definition the *input resistance* R. How does the input resistance vary with $x$? Part of the current flows toward the soma (proximal side), and the rest flows toward the distal axon. Thus, the input resistance decomposes into $R(x)^{-1} = R_{\text{proximal}}^{-1}(x) + R_{\text{distal}}^{-1}$, (for a long axon only the proximal resistance varies with $x$). The proximal segment is highly resistive because its end is sealed. Specifically, if $x$ is small ($x \ll \lambda$), which is the physiological situation when a current is injected at the distance of the AIS, we have $R_{\text{proximal}}(x)/R_{\text{distal}} \approx \lambda/x$, a large number (see Materials and methods). This means that the current flows mostly towards the distal axon (as seen in the uniform voltage response between soma and injection site, *Figure 1A*). In addition, the input resistance is approximately the distal resistance, and therefore the position of the injection site has little effect on the electrical response. More precisely, we can calculate that $R(x) \approx r_a(\lambda - x)$, where $r_a$ is axial resistance per unit length (*Figure 1B*). Thus, moving the AIS away from the soma should make the cell slightly less excitable. This is consistent with the findings of *Gulledge and Bravo (2016)*, who observed numerically that when the somatodendritic compartment is small, the neuron is most excitable when the AIS is next to the soma.

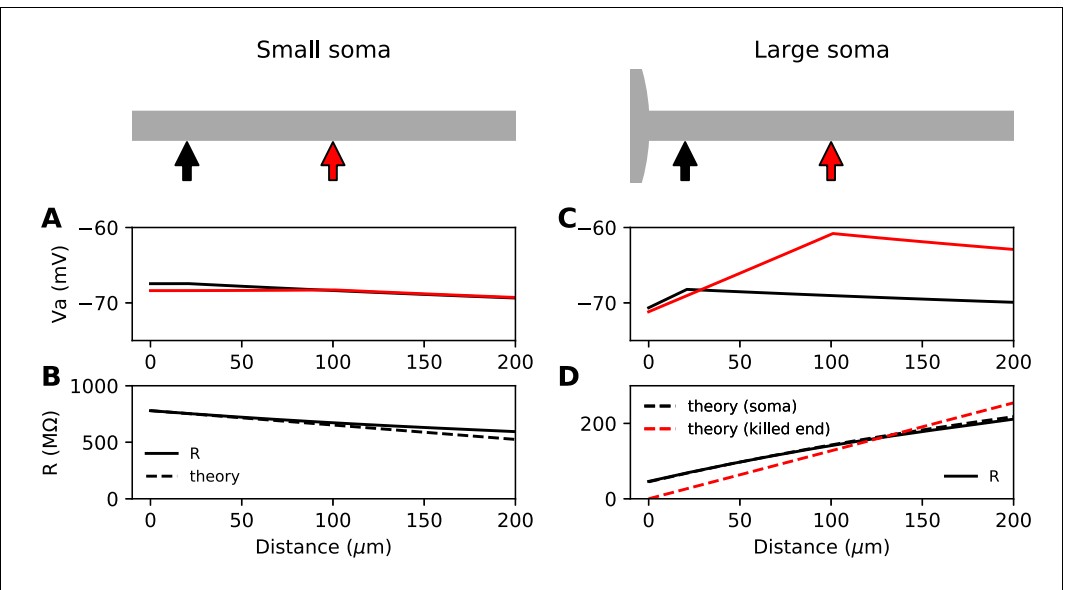

**Figure 1.** Steady-state passive response of the axon ($r_a$=1.3 MΩ/μm, λ=612 μm, $E_L$ = -75 mV). (A, B) With a very small soma; (C, D) With a large soma. (A) Voltage response along the axon for a 10 pA current injected at 20 μm (black) and at 100 μm (red). (B) Input resistance as a function of distance, as numerically measured (solid) and according to the simplified theoretical formula ($R(x) \approx r_a(\lambda - x)$, dashed). (C) Same as A for a large soma (diameter: 100 μm, current: 100 pA). (D) Same as B for a large soma, with the simplified theoretical prediction for the killed end condition ($R(x) \approx r_a x$, dashed red) and the full theoretical prediction for a finite soma ($R^{-1} = (r_a x + R_{\mathrm{soma}})^{-1} + (r_a \lambda)^{-1}$, dashed black; $R_{\mathrm{soma}}$ = 47.7 MΩ).

## Large soma, or killed end condition

Suppose now that the somatodendritic compartment is so large that the axial current has negligible effect on its potential. This is the 'killed end' condition (as if the membrane were open) (see Materials ans methods). In this case, when $x \ll \lambda$, $R_{\mathrm{proximal}}(x) \approx r_a x$, which means that only the axial resistive component is significant. We then have $R_{\mathrm{proximal}}(x)/R_{\mathrm{distal}} \approx x/\lambda$, a small number (the exact inverse of the sealed end condition). Thus, current flows primarily toward the soma and $R(x) \approx R_{\mathrm{proximal}}(x)$. We express this fact by stating that the soma is a *current sink*. This is illustrated on *Figure 1C* with a thin axon (diameter 1 μm) attached to a large spherical soma (100 μm). The current $I$ flowing toward the soma produces a linear depolarization between the soma and the injection site, with a total voltage difference $\Delta V = r_a x . I$. It follows that the input resistance increases with the distance of the injection site, which would tend to make the cell more excitable when the AIS is moved away from the soma, since less Na$^+$ current is then required to produce the same depolarization.

This simplified formula ($R(x) = r_a x$) differs from the actual input resistance in two ways, as illustrated in *Figure 1D*. First, some current flows toward the distal side, which becomes substantial at long distances from the soma. Second, with a finitely large soma instead of a killed end, the current also charges the soma, which makes the input resistance increase approximately by the somatic membrane resistance $R_{\mathrm{soma}}$. However, this difference holds for the stationary response. On a short time scale, the somatic depolarization is negligible because the soma charges much more slowly than the axon. This is illustrated in *Figure 2A*, where a current pulse is injected at the axon and measured at the injection site (red) and at the soma (black). The difference between the two responses, which is the voltage gradient between the soma and the injection site, essentially follows Ohm's law: $\Delta V = R_a I$ (*Figure 2B*), where $R_a = r_a x$ is the *axial resistance* of the axon between soma and injection site (thus, $R_{\mathrm{proximal}} \approx R_a + R_m$). Since on a short time scale the soma is not substantially depolarized, the local depolarization mainly reflects the ohmic voltage gradient across the proximal axon. In *Figure 2C*, we show the input resistance at time t = 300 μs, $(V_{\mathrm{axon}}(t = 300\ s) - V_0)/I$, as a function of distance $x$ (red): it is essentially the same as $\Delta V(t = 300\ s)/I$, because the somatic response

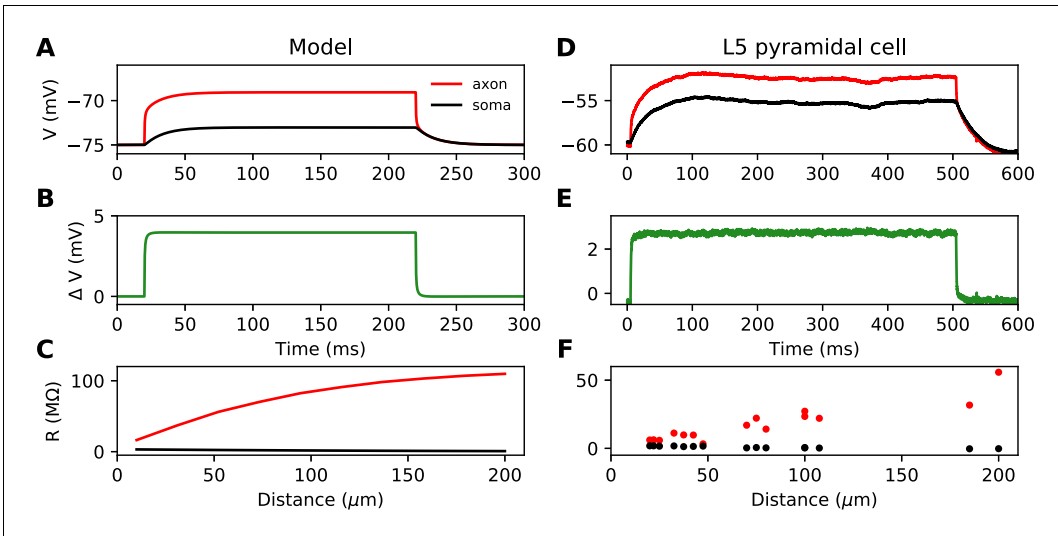

**Figure 2.** Time scale of responses to axonal current injection in a model with large soma (A-C) and in layer five pyramidal neurons (D-F). (A, D) Voltage response at the axonal injection site 75 µm away from the soma (red) and at the soma (black). (B, E) Difference between the two responses. (C, F) Input resistance measured 300 µs after the start of current injection as a function of the distance of the injection site (red), compared to the resistance obtained from the somatic voltage response at the same time (black).

$(V_\mathrm{soma}(t = 300\ s) - V_0)/I$ is negligible (black). This plot shows quantitatively that the axon indeed charges faster than the soma, for a broad range of distances of the axonal stimulation site.

To summarize, the case with a large soma is characterized by two key properties: 1) a current injected at the axon produces a resistive voltage gradient between the soma and injection site, proportional to the axial resistance $R_a$ of that piece of axon; 2) the axonal site charges much faster than the soma. We call this regime the *resistive coupling regime* (*Brette, 2013*; *Kole and Brette, 2018*; *Telenczuk et al., 2017*).

## Cortical cells

Which regime applies to neurons? Clearly, the soma of neurons is smaller than in the simulations shown in *Figure 2A–C*, but the axonal current must also charge the capacitance of the proximal dendrites (thus we call the compartment connected to the axon the somatodendritic compartment). We start by examining experimental recordings from layer 5 cortical pyramidal neurons, where a current pulse is injected in the axon and simultaneously recorded at the soma (*Hu and Bean, 2019*; *Hu and Bean, 2018*). These neurons have a soma of about 30 µm diameter attached to a large apical dendrite and an AIS of about 1–1.5 µm diameter (*Hamada et al., 2016*; *Höfflin et al., 2017*) – note that these are optical measurements, which have limited precision. Strictly speaking, the experimental situation is not exactly the same as the physiological situation because currents are injected in axonal blebs, and therefore the resistance of the distal axon is replaced by the resistance of the bleb. However, recordings of action potentials in intact AIS appear very similar to recordings in blebs (compare *Yu et al., 2008* with *Kole and Stuart, 2008*).

*Figure 2D* shows the response to a current pulse injected at 75 µm away from the soma, in the soma and at the axonal injection site. As in the theoretical case described above, a voltage gradient develops very quickly between the soma and injection site (*Figure 2E*). Note that the resting potential is different at the two sites; we will come back to this issue in a later section. As noted by *Hu and Bean (2018)*, the axonal input resistance increases with distance of the injection site. When measured 300 µs after the start of the pulse, the axonal input resistance increases steeply with distance, while the soma barely responds (*Figure 2F*). Therefore, the passive properties of these neurons follow the resistive coupling regime, rather than the small soma regime (compare with *Figure 1B*).

## Dimensional analysis

Could it be that large neurons follow the resistive coupling regime while smaller neurons such as granule cells do not? Consistently with this hypothesis, in simulations, *Gulledge and Bravo (2016)* noted that large neurons are more excitable when the AIS is distal whereas small neurons are more excitable when the AIS is proximal. However, those simulations were performed with a constant AIS diameter of 1.5 µm. As noted in *Telenczuk et al. (2017)*, cerebellar granule cells, which have a small cell body of about 6 µm diameter (*Delvendahl et al., 2015*) also have very thin axons, of diameter about 0.2 µm (*Perge et al., 2012*; *Wyatt et al., 2005*), and as a result they still follow resistive coupling theory because the soma remains large compared to the axon.

We now examine the relation between soma and axon diameter empirically. *Figure 3* shows minimum AIS diameter (generally measured at the distal end of the AIS) vs. soma diameter measured with electron microscopy in several cell types, plotted in logarithmic scale. We have excluded optical microscopy measurements because small AIS diameters approach the diffraction limit. The figure includes three sets of measurements on individual neurons: human spinal motoneurons (*Sasaki and Maruyama, 1992*), cat spinal motoneurons (*Conradi and Ronnevi, 1977*) and pyramidal and stellate cortical cells of primates (*Sloper and Powell, 1979*) (dots). It also includes average AIS and soma diameters of four other cell types: cat olivary cells (*Ruigrok et al., 1990*; *de Zeeuw et al., 1990*); rat CA3 pyramidal cells (*Buckmaster, 2012*; *Kosaka, 1980*); rat Purkinje cells (*Somogyi and Hámori, 1976*; *Takacs and Hamori, 1990*); mouse cerebellum granule cells (*Delvendahl et al., 2015*; *Palay and Chan-Palay, 2012*; *Wyatt et al., 2005*).

The data show that smaller neurons also tend to have a smaller AIS. The correlation between AIS and soma diameter appears both within and across cell types (the best power law fit across all merged data has exponent 1.14 ± 0.05, bootstrap standard deviation, but the exact number is not very meaningful because the regression is done on groups of different sizes).

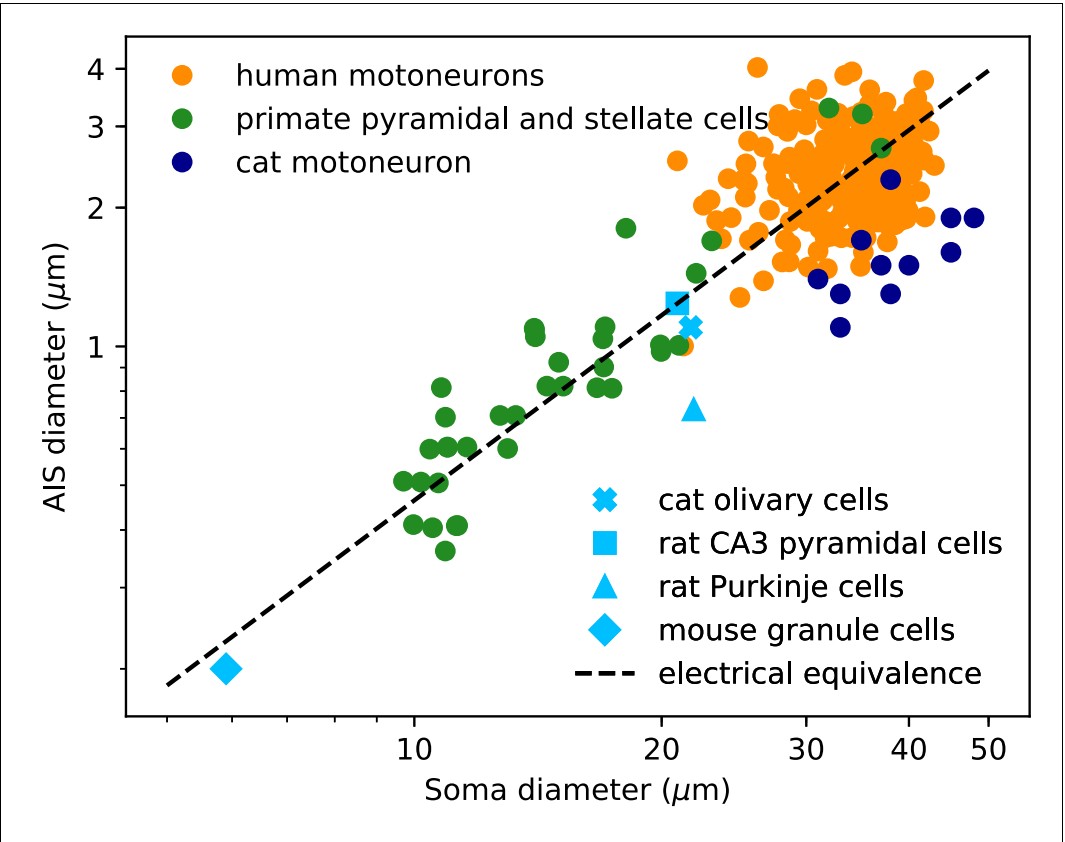

**Figure 3.** AIS diameter vs. soma diameter in a variety of cell types. Four points are averages over many neurons of the same type (light blue symbols), other points are individual measurements (dots). Electrical equivalence is represented by the dashed line.

To interpret this relation, we now ask what relation between soma and axon diameter preserves electrical properties. We use the dimensional analysis that *Rushton (1951)* applied to the scaling of conduction velocity with axon diameter. For this analysis, we consider a simplified model consisting of a spherical soma of diameter $d_S$ with no dendrites and a cylindrical axon of diameter $d_a$. First, the somatic input resistance scales as $R_s \propto d_s^{-2}$ (inverse membrane area), while the distal axonal resistance scales as $R_{\text{distal}} \propto d_a^{-3/2}$ (input resistance of a semi-infinite cylinder). For the resistance of the proximal axon, we must determine the injection point. To preserve electrical properties, its position should be constant in units of the space constant $\lambda$, which gives $R_a \propto \lambda/d_a^2 \propto d_a^{-3/2}$. Thus, for all resistances to scale in the same way and therefore to preserve electrical properties, we must have $d_s^{-2} \propto d_a^{-3/2}$, which means: $d_a \propto d_s^{4/3}$, that is, a power law exponent of about 1.3. This relation is represented by the dashed line on Figure 3: all lines parallel to it correspond to electrical equivalence – this is of course approximate since it does not take into account the scaling of dendrites.

In summary, small neurons also have a thin AIS, such that electrical properties are preserved. This indicates that the passive properties of the soma-AIS system should generically follow the resistive coupling regime. We will now show that in this regime, the appropriate measure of excitability for studying the effect of AIS structural plasticity is the somatic voltage threshold.

## Measuring excitability

### A simple biophysical model of spike initiation

We first present a minimal biophysical model of spike initiation that will be compared with theoretical predictions. We aimed for a simple model with as few parameters as possible (see Materials and methods for details). The morphology consists of a spherical soma, a large cylindrical dendrite and a thin cylindrical axon (*Figure 4*, top). The AIS is a section of the proximal axon with a high uniform density of inactivating sodium and non-inactivating potassium channels. The rest of the neuron contains a lower density of both channels. On this figure, the AIS is $L = 30$ μm long and is positioned at a distance $\Delta = 5$ μm from the soma.

For channels, we chose simple Hodgkin-Huxley-type models with just three interpretable parameters: half-activation (or inactivation) voltage $V_{1/2}$, slope $k$ and maximum time constant $\tau^{\max}$. The equilibrium value of gating variables is shown on *Figure 4A*. The activation slope factor of the Nav channel was rounded at 5 mV. We modeled the Kv channel in the same way, but we used 8 gates ($n^8$) as in *Hallermann et al. (2012)*. This was important so that the Kv channel activates with a delay. The dynamics of the gating variables during a spike are shown at the distal end of the AIS in *Figure 4B*.

Our goal was to reproduce the essential phenomenology of action potentials recorded in cortical neurons. First, a high amplitude action potential initiates first in the AIS where it rises quickly (dV/dt > 1,000 V/s), then appears in the soma with a distinct kink and a biphasic phase plot (*Figure 4C, D*; *Kole et al., 2008*; *Kole and Stuart, 2008*; *Naundorf et al., 2006*; *Stuart et al., 1997*; *Yu et al., 2008*). Second, $Na^+$ and $K^+$ currents have little overlap at spike initiation (*Figure 4E,F*; *Hallermann et al., 2012*), which implies that Kv channels are more involved in repolarization than in spike initiation.

### How to measure excitability?

Excitability changes associated with changes in AIS geometry have been reported using various measures: rheobase, the minimal constant current required to elicit a spike (*Lezmy et al., 2017*), or minimal transient current (*Raghuram et al., 2019*); threshold current density, which is rheobase divided by input capacitance (*Grubb and Burrone, 2010*; *Wefelmeyer et al., 2015*); somatic potential at spike onset (*Kuba et al., 2015*; *Kuba et al., 2014*).

All these quantities are related to each other. However, an issue with both rheobase and current density threshold is that they vary not only with changes in the AIS but also with the neuron's input resistance. For example, *Grubb and Burrone (2010)* reported an increase of about 50% in threshold current density after long-term depolarization associated with a distal displacement of the AIS. To infer the specific effect of changes in the AIS, they had to discount the estimated effect of an observed reduction of input resistance (about 1/3). This issue is illustrated in *Figure 5A and B*: the rheobase varies when the total Nav conductance $G$ is varied at the AIS, but also when the somatic

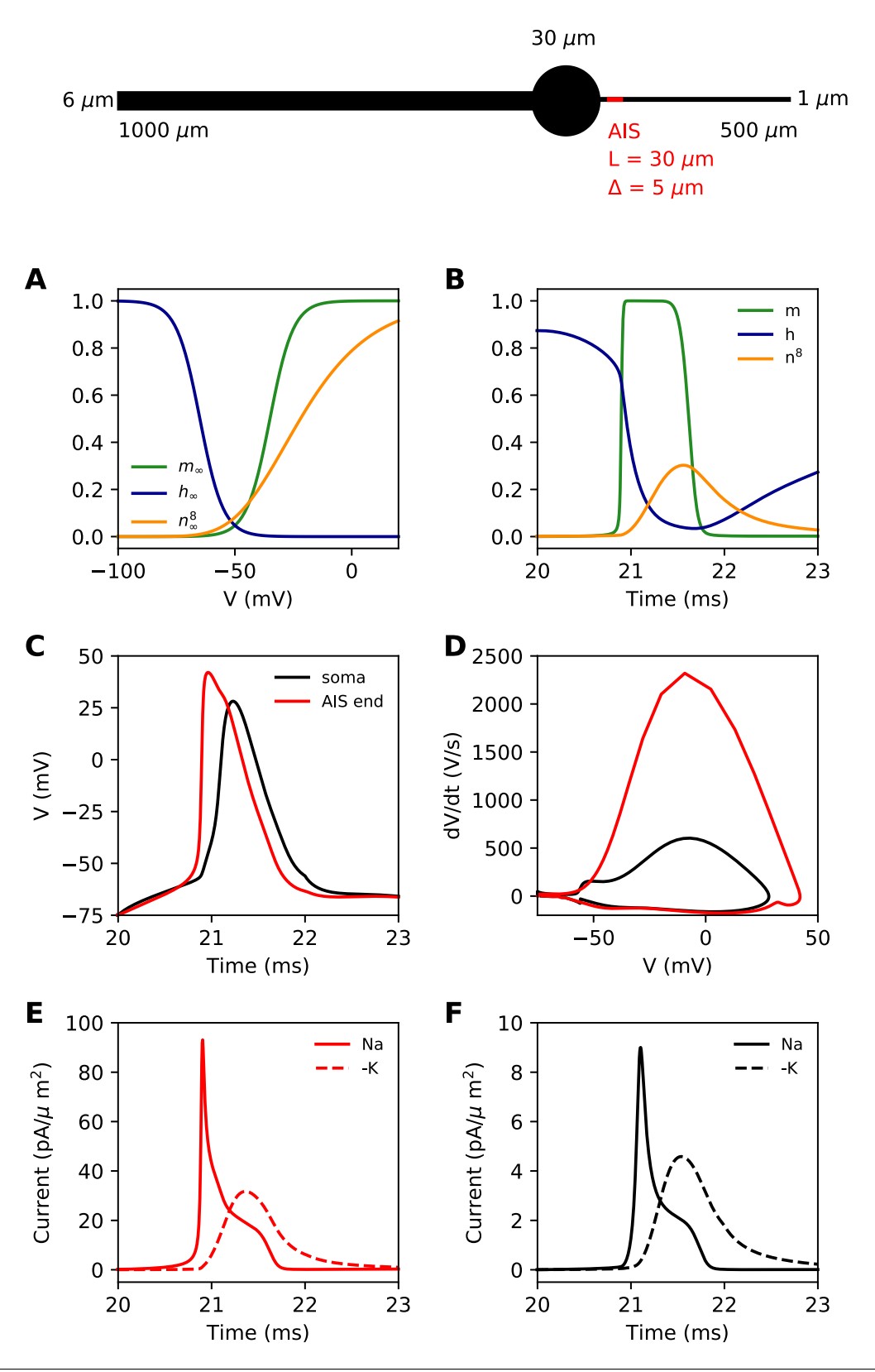

**Figure 4.** Simple biophysical model of spike initiation. Top, morphology of the neuron. (**A**) Equilibrium functions of the gating variables m, h, and $n^8$. (**B**) Time course of the gating variables at the distal end of the AIS during an

*Figure 4 continued on next page*

*Figure 4 continued*

action potential. (**C**) Action potential observed at the end of the AIS (red) and at the soma (black). (**D**) The same action potential shown in a phase plot. (**E**) Absolute value of Na$^+$ and K$^+$ currents at the AIS during a spike. (**F**) Same as E, at the soma.

leak conductance $g_L$ is varied. Thus, changes in rheobase reflect excitability changes due to factors in the AIS and elsewhere.

In the resistive coupling regime, the somatic voltage threshold is a measure of excitability that depends only on axonal factors. It is defined as the maximum somatic membrane potential that can be reached without triggering a spike (*Brette, 2013*). This is illustrated in *Figure 5A and B*: the voltage threshold varies with $G$ in the same way as the rheobase (more precisely, $V_{\text{threshold}} - V_0$ is proportional to the rheobase), but it does not vary with $g_L$. The reason is the separation of time scales between somatic and axonal dynamics, due to the size difference: when the Na$^+$ current enters the AIS, it depolarizes the AIS very rapidly, while at the time scale of spike initiation the soma does not get charged significantly. This is shown in *Figure 2C and F*, which compare the somatic and axonal responses at time t = 300 μs after current injection in the axon. It follows that at the relevant time scale, the somatic potential acts as a fixed boundary condition for the axon. Therefore, the initiation of a spike depends only on the somatic potential and properties of the axon, but not on properties of the soma or dendrites (an exception is the axon-carrying dendrite of some neurons, *Kole and Brette, 2018*). In terms of dynamical systems theory, the somatic potential is a bifurcation parameter (see below).

It could be argued that a better measure of excitability is the voltage threshold at the AIS, rather than at the soma, since spikes are initiated in the AIS (*Yu et al., 2008*). To show that this is not the case, we inject a constant hyperpolarizing current at the distal end of the AIS, while still triggering spikes with a somatic current (*Figure 5C*). As a result, the somatic voltage threshold is raised in proportion of the hyperpolarizing current. This change does represent a reduction of excitability, because the rheobase also increases - we note that the rheobase increases both because the voltage threshold increases and because the resting potential also decreases substantially (see final section). In contrast, the voltage threshold at the AIS changes in the opposite direction (*Figure 5C*): it does not capture the change in excitability due to this particular change at the AIS, but on the contrary it is misleading. We will analyze this perhaps counter-intuitive phenomenon in the last part of the manuscript. We note for now that the somatic voltage threshold specifically captures the axonal factors of excitability.

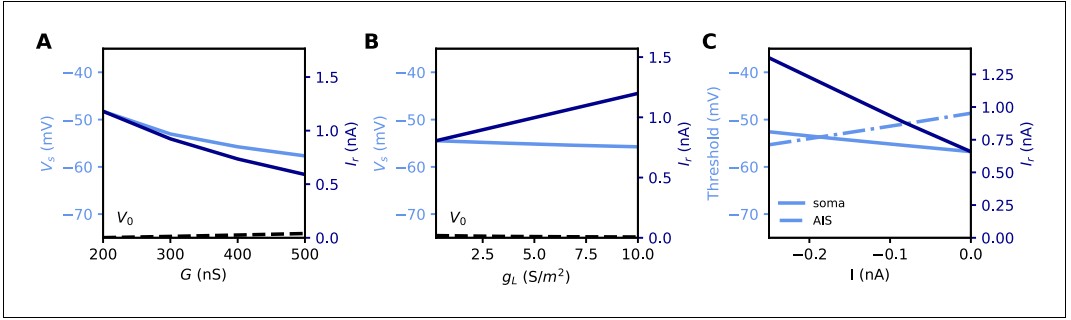

**Figure 5.** Measuring excitability in the biophysical model. (**A**) Rheobase (dark blue) and somatic voltage threshold (light blue) as a function of total Nav conductance in the AIS G (d$_S$ = 30 μm). The resting potential also changes slightly (dashed). (**B**) Rheobase and somatic voltage threshold as a function of leak conductance density (G = 350 nS). (**C**) Voltage threshold at the soma (solid light blue) and AIS (dashed light blue) as a function of a hyperpolarizing current injected at the AIS end. The rheobase is shown in dark blue.

## Relation between excitability and AIS geometry

### General theoretical framework

We now develop a theoretical analysis of the relation between AIS geometry and excitability as measured by the somatic voltage threshold. To develop the theory, we make a number of strong approximations. First, we neglect the axial current toward the distal axon because the soma acts as a current sink for the AIS. Second, we neglect axonal leak currents because the resistance through the membrane is much larger than the axial resistance toward the soma. For a length $x$ of axon, the membrane resistance is $R_m/(\pi d x)$ ($d$ is axon diameter and $R_m$ is specific membrane resistance); for example, with $d$ = 1 µm, $x$ = 100 µm and $R_m$ = 15 000 Ω.cm$^2$, we obtain about 4.8 GΩ. In contrast, with the same parameter values and $R_i$ = 100 Ω.cm we obtain an axial resistance $R_a = 4 R_i x/(\pi d^2) \approx 127$ MΩ. Third, we neglect the time-varying K$^+$ current responsible for repolarization because it is small compared to the Na$^+$ current at spike initiation, at least in layer 5 pyramidal cells (*Hallermann et al., 2012*) (see *Figure 4E*). Fourth, we further assume that Nav channel inactivation plays no role, except for setting the proportion of initially available Nav channels. Finally, we neglect all time-varying processes. This drastic approximation is justified by the following arguments: first, the somatic membrane potential should not vary substantially at the time scale of spike initiation because the somatodendritic compartment is large; second, the axonal capacitive current should be small because we are considering the situation near threshold (i.e. where $dV/dt$ is not very large) and the axonal capacitance is relatively small; third, the activation time constant of Nav channels is very short (about 100 µs at room temperature, *Schmidt-Hieber and Bischofberger, 2010*).

With this set of approximations, the theory then considers just two currents: the axial resistive current, and an instantaneous axonal Na$^+$ current (in the last part we will consider the effect of a static non-sodium current). Theoretical predictions will be compared with simulations in the biophysical model, which does not make these approximations. We address increasingly complex situations, starting with a point AIS.

### A point AIS

The idealized case where all Nav channels are clustered at a point AIS has been treated theoretically in *Brette (2013)*. We briefly summarize the result. When the somatic potential is increased, the axonal potential also increases. Under some condition, there is a somatic potential above which the axonal potential suddenly jumps to a higher value, which corresponds to spike initiation. This is called a bifurcation and the voltage at the bifurcation (the spike threshold) has been calculated analytically (see Materials and methods). A simple (but not rigorous) way to obtain the result is the following. The axial current is resistive, and therefore scales inversely with axial resistance $R_a$. At spike initiation, we therefore expect: $I_{\text{axial}} \propto 1/R_a$. The Na$^+$ current changes approximately exponentially with voltage below threshold (*Baranauskas and Martina, 2006*; *Hodgkin and Huxley, 1952b*): $I_{Na} \propto G \exp(V/k)$, where $G$ is the total available Na$^+$ conductance and $k$ is the Boltzmann slope factor of Nav channels (typically 4–8 mV *Angelino and Brenner, 2007*; *Platkiewicz and Brette, 2011*). The two currents must match (the Na$^+$ current entering the axon then flows toward the soma as a resistive current), therefore the somatic spike threshold is:

$$V_s = \text{constant} - k \log(R_a G)$$

where the constant depends on Nav channel properties. For an axon with constant section (e.g. cylindrical), the axial resistance $R_a$ is proportional to AIS position $\Delta$. Therefore, the formula can be expressed as

$$V_s = \text{constant} - k \log \Delta - k \log G$$

We did not include the AIS diameter $d$ in this formula, which would contribute an additional term $2k \log d$ (because $R_a$ is inversely proportional to d$^2$; see Discussion). *Figure 6A and B* illustrate this formula for $k$ = 5 mV. To show that the analysis is correct, we first show numerical results in a simplified cable model that includes neither Nav channel inactivation nor Kv channels (see Materials and methods). The soma is voltage-clamped and the command potential is increased until a spike is triggered in the AIS. This situation is close to the approximations used for the theory, but

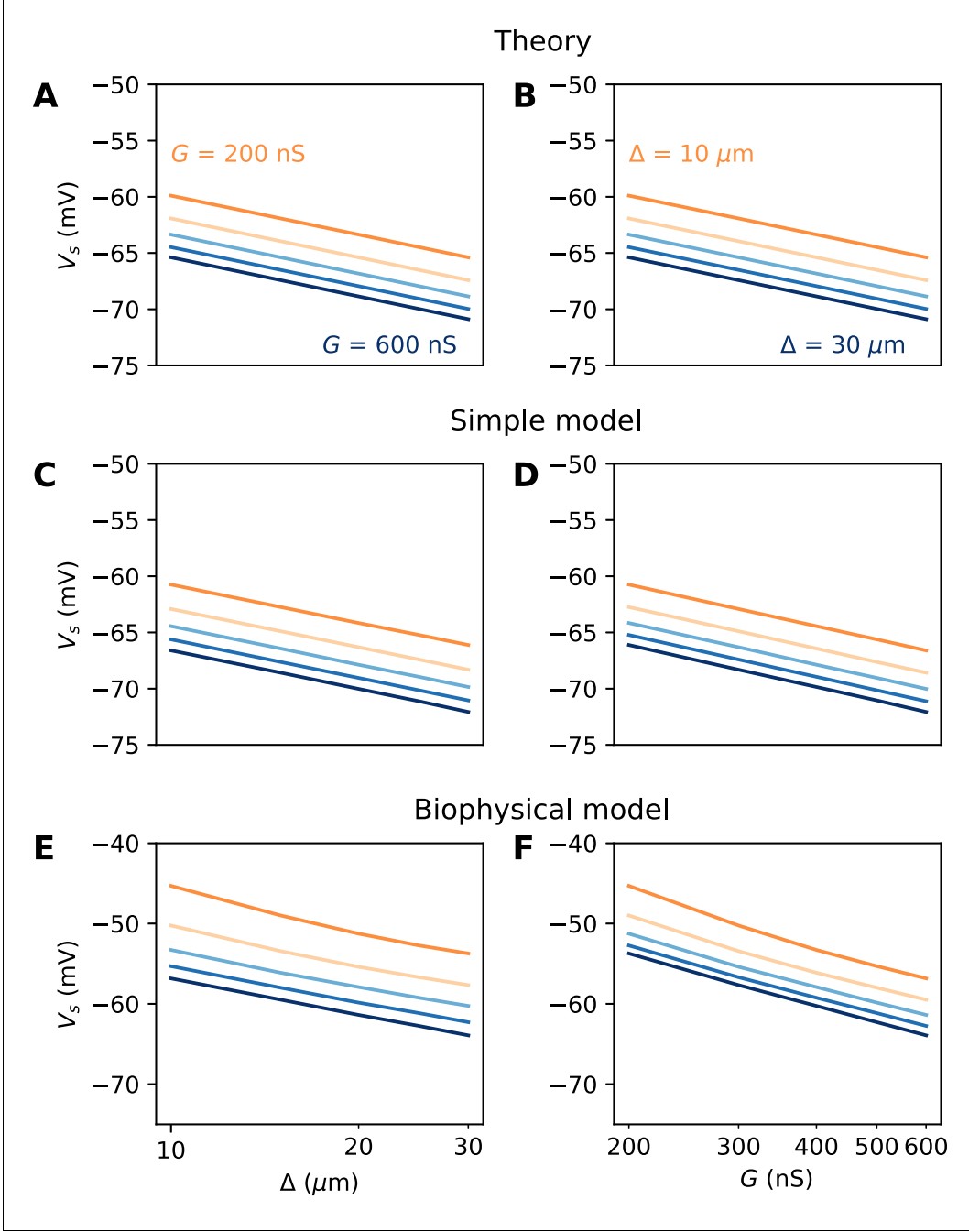

**Figure 6.** Spike threshold vs. AIS position and Nav conductance with a point AIS. (**A**) Theoretical prediction of spike threshold vs. AIS position in logarithmic scale, for different total Nav conductances (from 200 to 600 nS). (**B**) Theoretical prediction of spike threshold vs. total Nav conductance for different AIS positions (from 10 to 30 μm). (**C**) Spike threshold in a simplified model measured in somatic voltage clamp, vs. AIS position. The regression slope is about 5 mV for all curves. (**D**) Spike threshold in the simplified model vs. total Nav conductance. The regression slope varies between 5.3 mV (Δ = 10 μm) and 5.4 mV (Δ = 30 μm). (**E**) Spike threshold vs. AIS position in the biophysical model in current clamp. Regression slopes vary between 7.7 mV (G = 200 nS) and 6.5 mV (G = 600 nS). (**F**) Spike threshold vs. total Nav conductance in the biophysical model. Regression slopes vary between 10.5 mV (Δ = 10 μm) and 9.3 mV (Δ = 30 μm).

includes leak currents, axonal capacitive currents and Nav activation dynamics. *Figure 6C and D* show that the formula is essentially correct in this case.

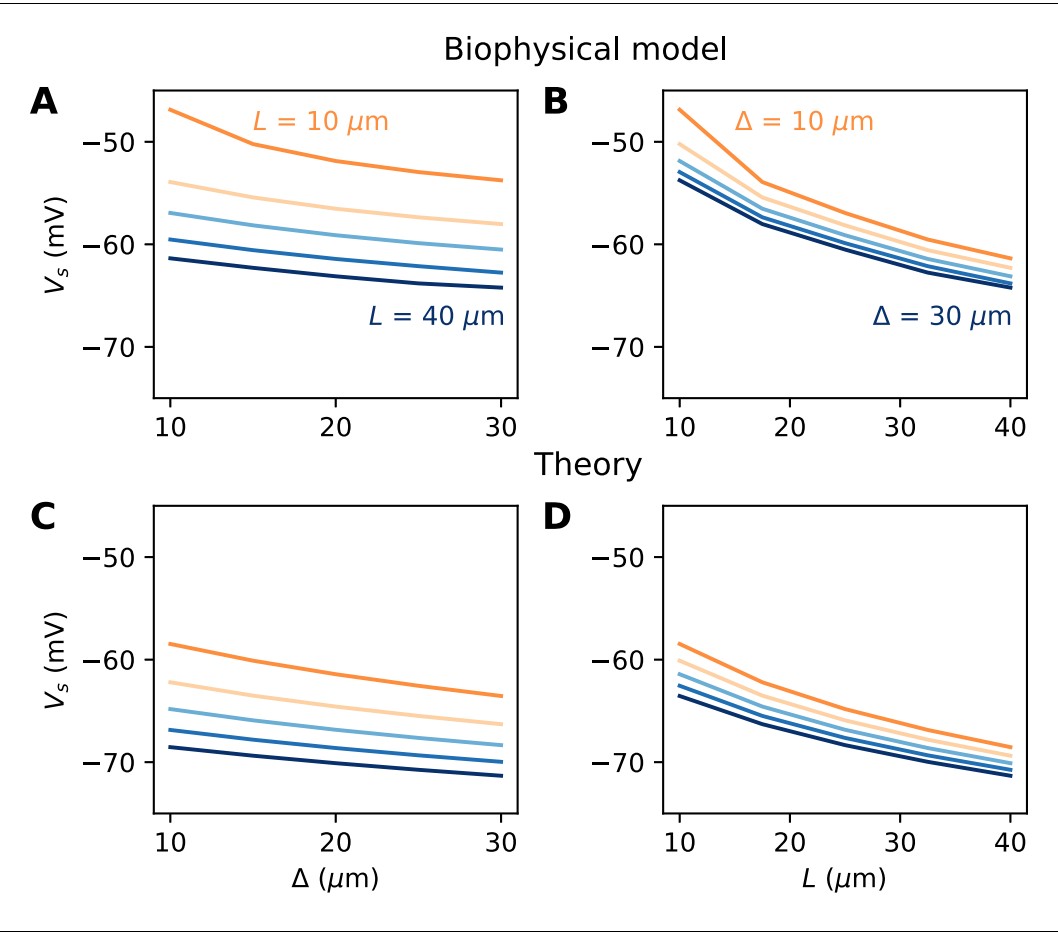

**Figure 7.** Relation between AIS geometry and voltage threshold in the biophysical model with constant Nav channel density. (**A**) Threshold vs. AIS position Δ (distance between soma and AIS start), for different AIS lengths between 10 and 40 μm. (**B**) Threshold vs. AIS length for different AIS positions between 10 and 30 μm. (**C**) Theoretical relation between threshold and AIS position (corresponding to panel **A**). (**D**) Theoretical relation between threshold and AIS length (corresponding to panel **B**).

We now examine this relation in the biophysical model where a spike is elicited in current-clamp. Several factors make this situation much more challenging: first, the somatic potential is no longer assumed to be clamped at spike initiation, and second, the model includes Nav channel inactivation and Kv channels, both of which can interfere with spike initiation. To ensure that the same proportion of Nav channels is initially available when $x$ and $G$ are varied, we use the following protocol: the somatic potential is held at $V_0 = -75$ mV, then a current step of varying amplitude is injected. The voltage threshold is defined as the maximal somatic potential reached in a non-spiking trial. Thus, the value of G represents the total non-inactivated conductance, which can potentially vary with the initial potential (see *Platkiewicz and Brette, 2011*; *Platkiewicz and Brette, 2010* for a theory); in practice, this variation was small in our simulations.

The numerical results show quantitative differences with the theoretical predictions (*Figure 6E and F*), namely, the spike threshold is more sensitive to Nav conductance than predicted (logarithmic slope of about 9 mV instead of $k = 5$ mV). Nonetheless, theory correctly predicts that 1) shifting the AIS distally or increasing total Nav conductance lowers spike threshold on a logarithmic scale, 2) the two logarithmic factors interact linearly (meaning the plotted lines are parallel). The relation between threshold and AIS position is also quantitatively well predicted (logarithmic slope of about 6–7 mV).

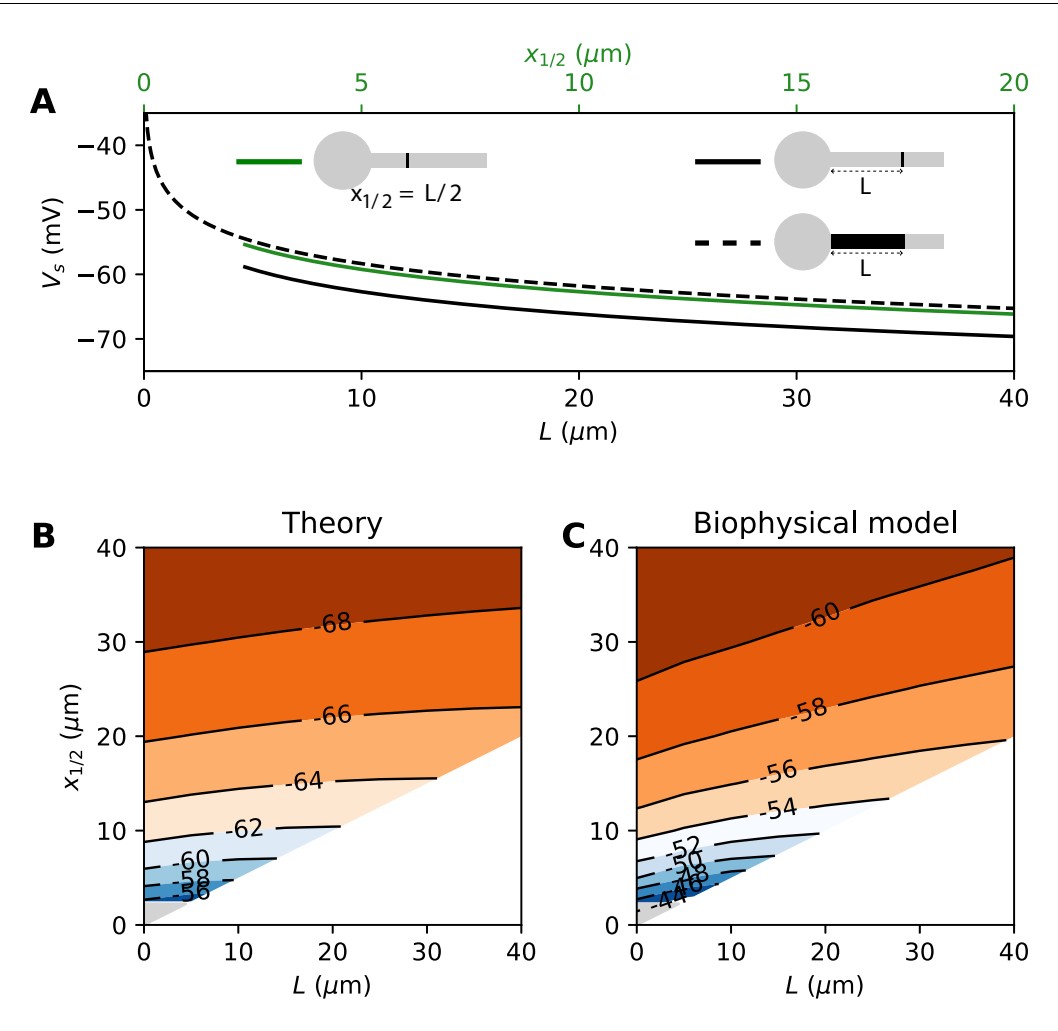

**Figure 8.** Effect of compressing the AIS on spike threshold, with total Nav conductance held fixed. (**A**) Theoretical spike threshold of an AIS of length L starting from the soma (dashed), compared to an equivalent point AIS placed at position Δ = L (solid black) and at Δ = $x_{1/2}$ = L/2 (solid green). (**B**) Theoretical spike threshold (in mV) vs. AIS middle position and length, with fixed total Nav conductance. The lower right white triangle corresponds to impossible geometries; the lower left shaded triangle is the region where the threshold is undefined in the point model. (**C**) Observed relation in the biophysical model.

## A spatially extended AIS

We now turn to the more difficult case of a spatially extended AIS, which requires new theory. *Figure 8* shows the relation between AIS geometry and voltage threshold in the biophysical model, when the surface conductance density of Nav channels is maintained constant. As expected, the threshold is lowered when the AIS is moved away from the soma (*Figure 7A*). However, the relation is less steep than in the case of a point AIS. The threshold is also lowered when the AIS is extended (*Figure 7B*). This is not surprising, but the quantitative relation cannot be easily extrapolated from the point AIS theory: when the AIS is extended, the total Nav conductance increases in proportion of the length, but the spike initiation site also moves.

We developed an analytical strategy to derive a formula for the extended case (illustrated in *Figure 7C,D*). We first consider an AIS of length *L* starting at the soma. We solve the cable equation in the resistive coupling regime, that is, where all time-varying effects as well as leak currents are neglected and the somatic potential is held fixed. The cable equation then becomes an ordinary second-order differential equation:

$$\frac{d^2V}{dx^2} \propto -ge^{V/k}$$

where $g$ is the surface conductance density of (available) Nav channels, and the proportionality factor includes axon diameter, resistivity and Nav channel properties. This equation simply expresses the fact that the $Na^+$ current entering the axon (right hand-side) equals the axial diffusion current (left hand-side). This equation can be solved with the two boundary conditions: $V(0)$ is the somatic potential ($V(0) = V_s$) and no axial current flows toward the distal axon ($V'(L) = 0$; see Materials and methods). A simple rescaling argument shows that the spike threshold varies as

$$V_s = \text{constant} - k \log G - k \log L$$

where $G$ is the total Nav conductance ($G = \pi dL$ for a cylindrical axon). Remarkably, this is the same formula as for a point AIS placed at position $\Delta = L$ and with the same total number of Nav channels. The only difference is the constant term. By solving the differential equation analytically, we find that this difference is *0.87 k* (about 4.3 mV with $k$ = 5 mV), as illustrated on *Figure 8A* (dashed vs. solid black). Note that in the point model, the threshold is not defined anymore (no bifurcation or 'kink') when AIS distance is lower than 2.25 μm (*Brette, 2013*). This limit is more difficult to calculate in the extended model.

An equivalent and more intuitive way to understand this result is to note that a threshold shift of *0.87 k* is equivalent to a displacement of the AIS by a factor $e^{-0.87} \approx 0.42$. In other words, the spike threshold of the extended AIS is almost the same as for a point AIS with the same number of channels placed in the middle of the AIS (*Figure 8A*, green curve):

$$V_s = \text{constant} - k \log G - k \log x_{1/2}$$

with $x_{1/2}$ = L/2, the midpoint of the AIS, and the constant in this formula differs from the constant for the point AIS by about 0.9 mV (see Materials and methods). This analysis shows that extending the AIS lowers the spike threshold by two mechanisms: by increasing the number of Nav channels and by moving the initiation site away from the soma. As $G$ and $x_{1/2}$ are both proportional to $L$, each factor contributes a shift of $-k \log L$.

We can now consider the general case of an extended AIS of length $L$, placed at distance $\Delta$ from the soma. The exact same analytical strategy can be applied, the only difference being the boundary condition at the start of the AIS, which now expresses the fact that the current flowing toward the soma is proportional to the voltage gradient between soma and AIS according to Ohm's law. This can be solved analytically (see Materials and methods), and we find that the spike threshold is almost the same as if the AIS were compressed into a single point at its center ($x_{1/2} = \Delta + L/2$), up to a corrective term $kf(\Delta/L)$. With $k$ = 5 mV, this corrective term is at most 0.9 mV (see Materials and methods). *Figure 8B* shows that compressing or extending the AIS around its middle position $x_{1/2}$ without changing the total Nav conductance has very little effect on the theoretical spike threshold. In the simulated biophysical model, the effect is more significant but remains small (*Figure 8C*). Therefore, we find that with an extended AIS, the spike threshold is approximately the same as that of the equivalent point AIS placed at the middle position $x_{1/2}$, with the same total Nav conductance.

In summary, we have found a simple theoretical relationship between spike threshold and AIS geometry, as well as Nav conductance density:

$$V_s = \text{constant} - k \log g - k \log L - k \log x_{1/2}$$

The constant term captures the effects of Nav channel properties, intracellular resistivity and axon diameter (see Discussion). The remarkable finding is that the variation of spike threshold can be separated into three independent contributions. *Figure 9* indicates that this theoretical finding is essentially valid in the biophysical model. The relation of spike threshold with any of these three factors ($g$, $L$, $x_{1/2}$) is essentially the same logarithmic relation when the other two factors are changed, up to a constant shift (i.e. the relations appear as parallel lines). The observed slopes are close to the theoretical prediction $k$ = 5 mV for the geometric factors $x_{1/2}$ and $L$ (6–7 mV), and a little larger for $g$ (8–9 mV).

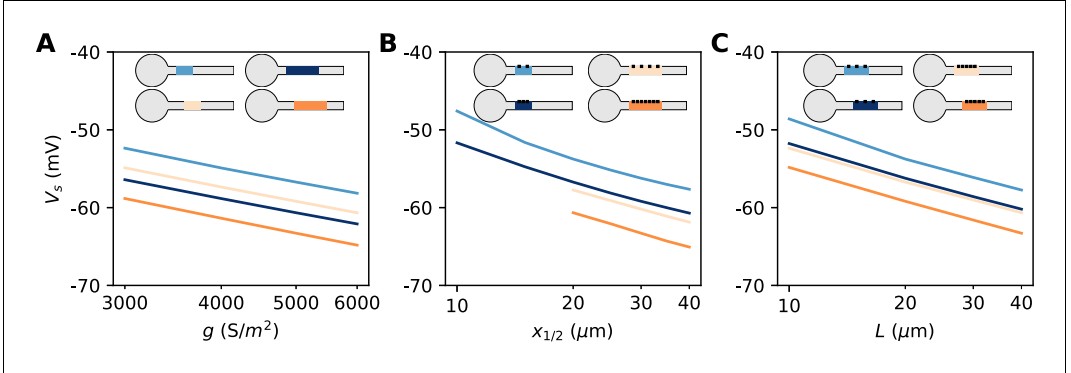

**Figure 9.** Dependence of spike threshold on AIS geometry and Nav conductance density in the biophysical model. (A) Spike threshold vs. Nav conductance density g, for 4 combinations of AIS middle position $x_{1/2}$ and length L (light blue, $x_{1/2}$ = 20 µm, L = 20 µm, regression slope: 8.4 mV; light orange, $x_{1/2}$ = 20 µm, L = 40 µm, regression slope: 8.3 mV; dark blue, $x_{1/2}$ = 30 µm, L = 20 µm, regression slope: 8.2 mV; dark orange, $x_{1/2}$ = 30 µm, L = 40 µm, regression slope: 8.7 mV). (B) Spike threshold vs $x_{1/2}$ for 4 combinations of g and L (light blue, g = 3500 S/m$^2$, L = 20 µm, regression slope: 7.1 mV; light orange, g = 3500 S/m$^2$, L = 40 µm, regression slope: 6 mV; dark blue, g = 5000 S/m$^2$, L = 20 µm, regression slope: 6.5 mV; dark orange, g = 5000 S/m$^2$, L = 40 µm, regression slope: 6.4 mV). (C) Spike threshold vs. L for 4 combinations of g and $x_{1/2}$ (light blue, g = 3500 S/m$^2$, $x_{1/2}$ = 20 µm, regression slope: 6.6 mV; light orange, g = 5000 S/m$^2$, $x_{1/2}$ = 20 µm, regression slope: 6 mV; dark blue, g = 3500 S/m$^2$, $x_{1/2}$ = 30 µm, regression slope: 6.1 mV; dark orange, g = 5000 S/m$^2$, $x_{1/2}$ = 30 µm, regression slope: 6.1 mV).

We are now in better position to understand *Figure 7*. We noted that the spike threshold changes less than expected when the AIS is moved away. This is because it varies logarithmically with the middle position $x_{1/2}$ and not with the start position Δ. For example, for an AIS of length L = 40 µm placed at position Δ = 10 µm, a displacement of 5 µm shifts the middle position from 25 to 30 µm. The theory then predicts that the threshold decreases by $k \log(30/25) \approx 0.9$ mV (assuming $k$ = 5 mV), close the measurement in the biophysical model ($\approx$ 1.2 mV). Thus, the theoretical effect of AIS start position on excitability is significant but moderate.

If we extend the AIS while keeping the same start position, then the spike threshold decreases because of the increase in length (as $-k \log L$) and because the AIS middle position moves away (as

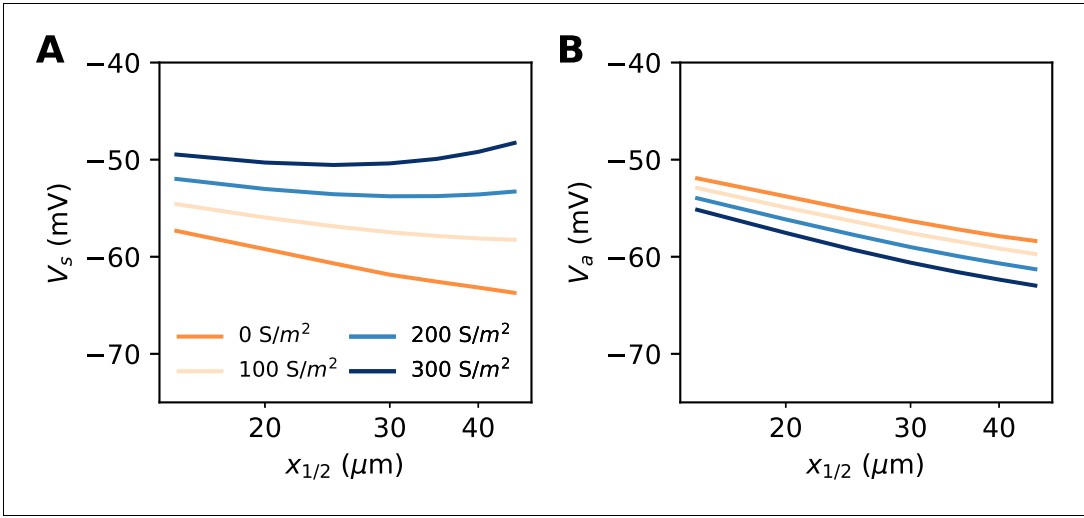

**Figure 10.** Excitability as a function AIS position with hyperpolarizing conductance (biophysical model). The conductance has reversal potential E = −90 mV and is uniformly expressed in the distal half of the 30 µm long AIS (G = 500 nS). (A) Somatic threshold vs. AIS position for different conductance values (an empirical estimate in layer 5 pyramidal cells is 144 S/m$^2$, *Battefeld et al., 2014*). (B) Threshold at the distal end of the AIS for different conductance values (logarithmic regression slopes: 5.9 to 7.1 mV).

$-k\log(\Delta + L/2))$. If the AIS is close to the soma, then these terms add up to $-2k\log L$. For example, an extension from 40 to 50 µm theoretically lowers the threshold by about 2.2 mV. Thus, for realistic changes in position and length, the theory predicts moderate changes in threshold, consistently with empirical observations in structural plasticity studies (see Discussion).

## Non-sodium axonal currents

So far, our theoretical analysis predicts that the neuron should be more excitable when the AIS moves away from the soma. Yet the opposite effect has been seen in some model simulations (*Grubb and Burrone, 2010*; *Lezmy et al., 2017*). Specifically, *Lezmy et al. (2017)* observed in a biophysical model that when a strong Kv7 conductance is placed along the AIS, moving the AIS away from the soma makes the neuron less excitable. This effect can be observed in our model if we add a strong hyperpolarizing conductance on the distal half of the AIS (*Figure 10A*). It contradicts our previous findings, which were based on an analysis of the Nav channels only. However, it appears that the voltage threshold measured at distal end of the AIS still follows the theoretical prediction, with spikes initiating at lower axonal voltage when the AIS is moved away from the soma (*Figure 10B*). We now analyze this phenomenon.

Let us consider a point AIS placed at position $\Delta$, in which a hyperpolarizing current $I$ is injected ($I<0$). This current could model Kv7 channels or synapses onto the AIS (*Wefelmeyer et al., 2015*). By resistive coupling, the current will hyperpolarize the AIS relative to the soma by an amount $\Delta V = R_a I = r_a \Delta I$ ($r_a$ is the axial resistance per unit length) (*Figure 11A*). As shown on *Figure 11B*,

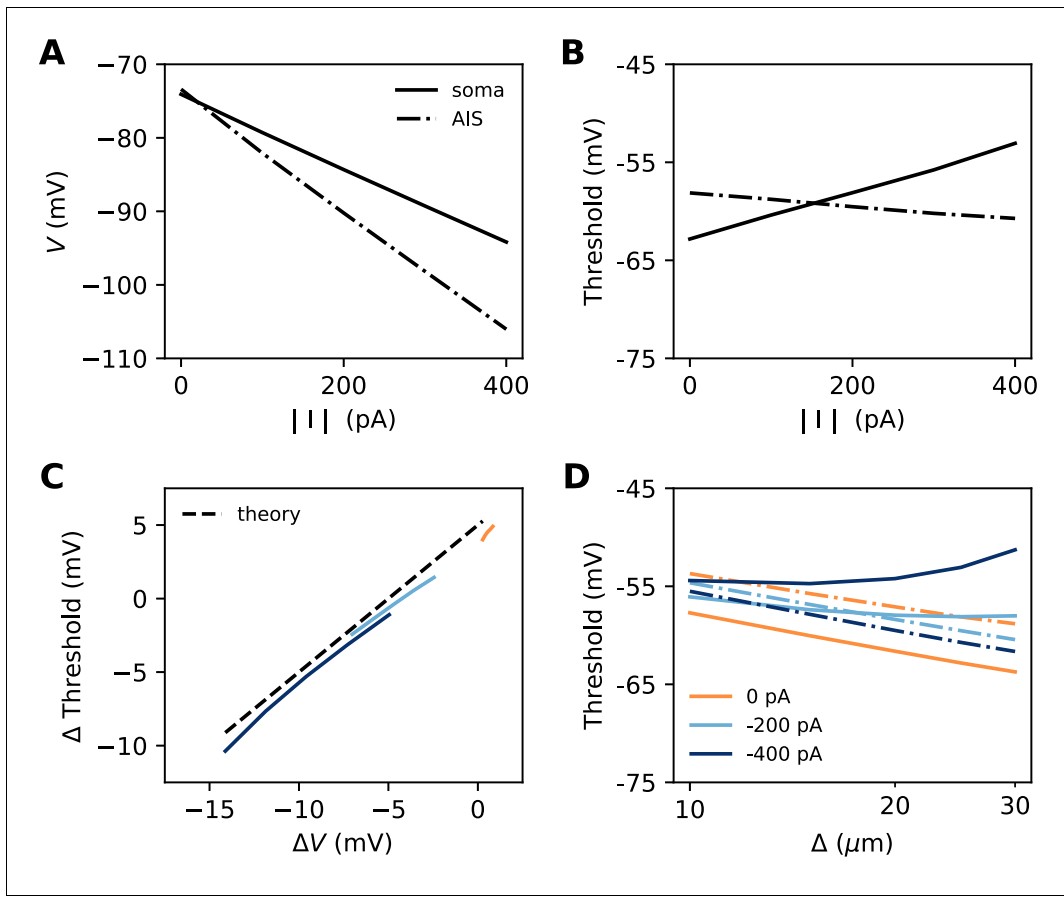

**Figure 11.** Effect of a hyperpolarizing axonal current in the biophysical model with a point AIS. (A) Resting potential at the soma and AIS as a function of current intensity |I|, for an AIS placed 25 µm away from the soma. (B) Threshold at the AIS and soma as a function of |I|. (C) Difference between AIS and somatic threshold vs. difference between AIS and somatic resting potential. Each curve corresponds to a variable AIS position with a given current intensity (0 to -400 pA). (D) Somatic (solid) and AIS threshold (dashed) vs. $\Delta$ when currents are injected at the AIS.

the voltage threshold at the AIS is not substantially affected by this current (see Materials and methods) – although a hyperpolarizing current may indirectly lower the threshold by deinactivating Nav channels (there is indeed a slight decrease with increasing current amplitude). It follows that the somatic threshold increases linearly with the current, by an amount equal to $\Delta V$. In the absence of hyperpolarizing current, the somatic threshold is theoretically below the AIS threshold by an amount $k$ (about 5 mV). The difference between somatic and AIS threshold is therefore predicted to be $-k - \Delta V$. This prediction holds in the biophysical model, as shown in *Figure 11C*, which compares the difference between somatic and AIS threshold with the difference between somatic and AIS resting potentials, when intensity and AIS position are varied.

Thus, when the AIS is moved away from the soma, the spike threshold at the AIS decreases as predicted in the previous section, but the somatic threshold differs from it by $-k - \Delta V$, which can make it increase with distance (*Figure 11D*).

Is this effect likely to be substantial in neurons? Empirically, *Hu and Bean (2018)* found that in layer 5 pyramidal cells, the AIS (more precisely, the axonal bleb) is about 3 mV hyperpolarized relative to the soma. This suggests that the contribution of this effect to threshold variations should be small. For example, suppose that a 45 µm AIS starting 10 µm away from the soma has distal Kv7 channels that hyperpolarize the AIS end by 3 mV. Then moving the AIS away from the soma by 10 µm (a large displacement) would move the AIS end from 55 to 65 µm, producing an increase in somatic threshold of about 0.5 mV by the effect discussed in this section (10/55 × 3 mV) – assuming the axonal resting potential is not homeostatically regulated. This increase in threshold would be more than compensated by the decrease due to the displacement of the Nav channels, of about 1.3 mV (using $k$ = 5 mV).

## Role of axon morphology

In the previous analysis, we have neglected the role of the distal axon. We now examine the impact of axon morphology on the results.

### Axon diameter

In the previous simulations, the biophysical models considered a 500 µm long unmyelinated axon of diameter 1 µm. Some neurons can have a much larger AIS: for example, many motoneurons shown in *Figure 3* have an AIS diameter of about 3 µm. This is not expected to change the electrical situation because, as that figure shows, the size of the soma scales with the size of the AIS. In *Figure 12A*, we simulated a biophysical model with an axon diameter $d_{AIS}$ = 3 µm and a soma scaled according to the power law: $d_S = d_{AIS}^{3/4}$. All other aspects of the neuron morphology are unchanged. The axonal space constant is increased by a factor $\sqrt{3}$, and therefore we scaled AIS position and length by the same factor. We can see that the spike threshold still varies with AIS position as expected (logarithmic regression slope: 6.8 mV in the original neuron vs. 7.1 mV in the large neuron).

In our theoretical analysis, we have assumed that the AIS diameter $d$ is fixed, but it is possible to take diameter changes into account in our analysis. Diameter contributes in two ways. First, axial resistance scales inversely with the axon section area, that is, $r_a \propto 1/d^2$. This contributes an additional term $2k \log d$ to the threshold. Second, for a fixed surface conductance density, the total conductance scales with axon diameter, $G \propto d$. This contributes an additional term $-k \log d$ to the threshold. Therefore, the extended formula reads:

$$V_s = \text{constant} - k \log x_{\frac{1}{2}} - k \log L - k \log g + k \log d$$

where we neglected the modulation by axonal currents. Thus, for a given AIS position, scaling AIS diameter by a factor 3 and AIS length by a factor $\sqrt{3}$ results in a positive shift of the threshold by $(k/2) \log 3 \approx$ 2.7 mV. This expected shift is shown on *Figure 12A* (dashed).

### Axon length and myelin

In our derivation of the spike threshold, we have ignored the distal axon, that is, we have considered that the resistance towards the distal axon is infinite (very large compared to the resistance towards the soma). More accurately, this resistance is large but finite and depends on the properties of the

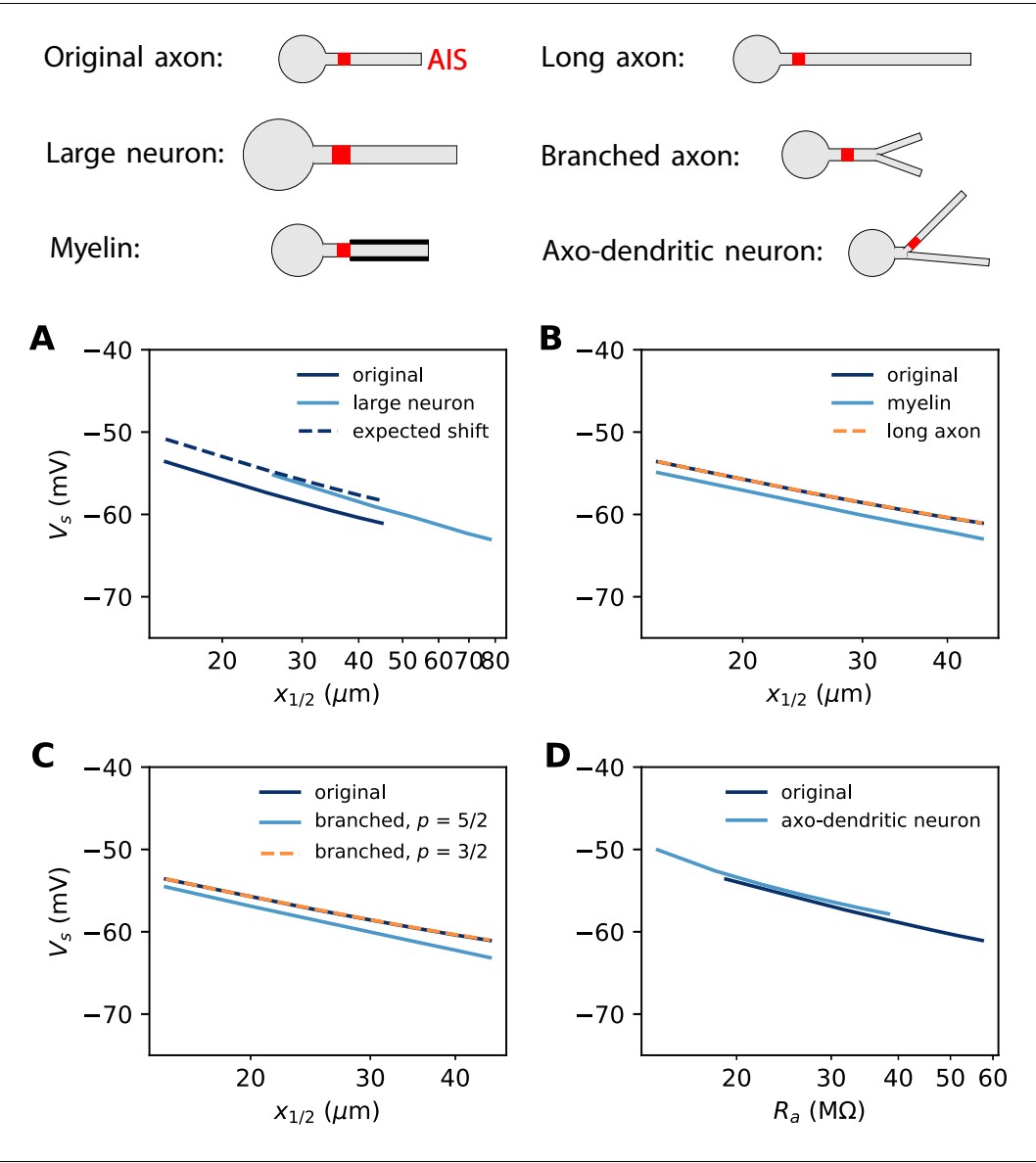

**Figure 12.** Effect of axon morphology on the relation between AIS position and excitability (biophysical model). Top: schematics of the different morphologies considered. (**A**) Somatic threshold vs. AIS position for a large neuron (light blue; diameter: 3 μm) compared to the original neuron (dark blue; diameter: 1 μm). AIS length is scaled up as the space constant, that is, by a factor $\sqrt{3}$. This scaling results in a theoretical threshold shift of about 2.7 mV (dashed). (**B**) Somatic threshold vs. AIS position for a longer axon (dashed orange) and for a myelinated axon (light blue), compared to the original neuron (dark blue). (**C**) Somatic threshold vs. AIS position for a branching axon with diameters such that $d_{\text{main}}^p = d_1^p + d_2^p$, where p = 5/2 (light blue) or 3/2 (dashed orange). (**D**) Somatic threshold vs axial resistance $R_a$ between soma and AIS middle position, in an axo-dendritic neuron (light blue). The axon-carrying dendrite starts with diameter 2 μm from the soma and splits after 7 μm into two branches of equal diameter (such that $d_{\text{main}}^{3/2} = d_1^{3/2} + d_2^{3/2}$).

distal axon. If the 500 μm axon is extended to a 1000 μm axon, then the distal axonal resistance $R_{\text{distal}}$ is barely affected and therefore the spike threshold does not change (***Figure 12B***). However, if the axon is myelinated, so that the specific membrane resistance is increased by a factor 4 (and capacitance decreased by the same factor), then the spike threshold is slightly lowered (***Figure 12B***, dashed).

To understand this, we can observe that the electrical impact of the distal axon is formally equivalent to a conductance $1/R_{\text{distal}}$ applied at the AIS end, with reversal potential equal to the resting potential (neglecting time-dependent effects). Therefore, we can apply the results of the previous section (*Non-sodium axonal currents*). The effect of a finite distal axonal resistance is thus to increase the somatic spike threshold by about:

$$\Delta V = \frac{R_a}{R_{\text{distal}}}(V_a - E_L)$$

where $V_a$ is the threshold at the AIS and $R_a$ is measured at the AIS end. As $R_{\text{distal}} = r_a\lambda$, we have $R_{\text{distal}} \propto \sqrt{r_m}$ where $r_m$ is the specific membrane resistance. Therefore, if the axon is myelinated, then this threshold shift should be halved, which implies that the spike threshold should be lowered. We can make a rough estimate in the case shown on *Figure 12B*. With the model parameters (*Table 1*), we obtain $R_{\text{distal}} \approx 780$ M$\Omega$, and $R_a \approx 95$ M$\Omega$ at the end of a 50 µm long AIS placed at position $x_{1/2}$ = 25 µm. With $V_a \approx -55$ mV, we obtain $\Delta V \approx 2.4$ mV for an infinite unmyelinated axon, With myelination, this shift is halved: $\Delta V \approx 1.2$ mV, and therefore myelination should lower the threshold by about 1.2 mV (this is an imprecise estimate since the axon is not infinite). In the simulation (*Figure 12B*), this shift was 1.3 – 1.9 mV. In summary, the effect of myelination on spike threshold should be limited.

**Table 1.** Parameters values of the biophysical model.
Time constants corrected for temperature are indicated in brackets.

| | | |
|---|---|---|
| Passive properties | $R_m$ | 15 000 $\Omega$.cm$^2$ |
| | $E_L$ | −75 mV |
| | $R_i$ | 100 $\Omega$.cm |
| | $C_m$ | 0.9 µF/cm$^2$ |
| Nav channels | $g_{Na}$, soma | 250 S/m$^2$ |
| | $g_{Na}$, dendrite and axon (non AIS) | 50 S/m$^2$ |
| | $g_{Na}$, AIS | variable (default: 3500 S/m$^2$) |
| | $E_{Na}$ | 70 mV |
| | $V_m^{1/2}$, soma | −30 mV |
| | $V_h^{1/2}$, soma | −60 mV |
| | $V_m^{1/2}$, AIS | −35 mV |
| | $V_h^{1/2}$, AIS | −65 mV |
| | $k_m$ | 5 mV |
| | $k_h$ | 5 mV |
| | $\tau_m^*$ | 150 µs (corrected: 54 µs) |
| | $\tau_h^*$ | 5 ms (corrected: 1.8 ms) |
| Kv1 channels | $g_K$, soma | 250 S/m$^2$ |
| | $g_K$, dendrite and axon (non AIS) | 50 S/m$^2$ |
| | $g_K$, AIS | 1500 S/m$^2$ |
| | $E_K$ | −90 mV |
| | $V_n^{1/2}$ | −70 mV |
| | $k_n$ | 20 mV |
| | $\tau_n^*$ | 1 ms |
| Kv7 channels | $g_{Kv7}$ | variable |
| | $E_K$ | −90 mV |

## Axon branching

The axon may also display complex branching patterns (*Wang et al., 2019*). The impact of branching on passive electrical properties is well understood from Rall's theory (*Rall, 2011*). In particular, Rall has shown that an axonal tree is electrically equivalent to an unbranched axon if diameters follow the rule $d_{\text{parent}}^{3/2} = d_{\text{daughter1}}^{3/2} + d_{\text{daughter2}}^{3/2}$. *Figure 12C* (dashed) shows that indeed in this case the spike threshold is unchanged. Empirically, this relation is not always satisfied: *Chklovskii and Stepanyants (2003)* report an average exponent of about 5/2, with large variability. In this case, the spike threshold is slightly lowered (*Figure 12C*, light blue). This shift can be estimated as previously, by calculating $R_{\text{distal}}$ with standard cable theory.

Branching can also occur before the AIS, when the AIS sits on an axon-bearing dendrite (*Kole and Brette, 2018*). In this case, since the diameter is not uniform, one must consider not metric distances but axial resistances: $x_{\frac{1}{2}}$ is then replaced by $R_a\left(x_{\frac{1}{2}}\right)$ in the equation, where the latter term is the axial resistance between soma and the AIS midpoint. *Figure 12D* shows the spike threshold as a function of axial resistance between the soma and AIS, for a neuron with a dendrite of 2 μm diameter that splits into two equal branches (according to Rall's law), one of which carrying the AIS. The spike threshold is almost unchanged compared to the original unbranched axon, for a given axial resistance.

Overall, axon morphology appears to have a small impact on excitability.

## Relation with experimental observations

We now discuss previous experimental observations in the light of this theoretical work. Not many studies have simultaneously reported changes in voltage threshold (rheobase is often reported instead) and in AIS geometry (position and length). First of all, it is important to stress that those observations cannot be used to directly validate or invalidate the theory, since many other relevant cell properties might also vary between groups (*Kole and Brette, 2018*), such as axon diameter; density, phosphorylation and inactivation state of Nav channels; expression level of Kv channels. Rather, we will provide theoretical changes in voltage threshold expected from observed AIS geometry changes, all else being equal (see *Table 2*). We will then discuss the discrepancy with observed threshold changes, in particular in terms of the unobserved factors.

We start with the single study of structural AIS plasticity in adult neurons that also reports voltage threshold. In neurons of chick nucleus magnocellularis, the AIS elongates by about 10 μm after 7 days of auditory deprivation, with no significant change in AIS start position (*Kuba et al., 2010*). It was reported that immunofluorescence intensity did not change, which suggests that at least the

**Table 2.** Changes in AIS geometry and voltage threshold ($\Delta V_s$) in structural plasticity and development studies, with the theoretical expectation, assuming constant functional Nav channel density and everything else unchanged (e.g. axon diameter, channel properties).

| Neuron type | Initial AIS L (μm) | Initial AIS $x_{1/2}$ (μm) | Final AIS L (μm) | Final AIS $x_{1/2}$ (μm) | $\Delta V_s$ (mV) | $\Delta V_s$ theory (mV) | Reference |
|---|---|---|---|---|---|---|---|
| *Plasticity* | | | | | | | |
| Chicks nucleus magnocelluaris | 9.6 | 13.3 | 19.5 | 18.4 | -4 | −5.2 | *Kuba et al., 2010* |
| Hippocampal cultures (only excitatory) | 34.8 | 20.9 | 33.6 | 27.2 | 4.3 | −1.1 | *Grubb and Burrone (2010)* |
| Hippocampal dentate granule cells in cultures | 19.2 | 10.4 | 15.7 | 7.85 | −1.1 | 2.4 | *Evans et al., 2015* |
| Olfactory bulb dopaminergic neurons in cultures | 11.7 | 21.1 | 14.2 | 15.5 | −0.4 | 0.6 | *Chand et al., 2015* |
| *Development** | | | | | | | |
| Chicks nucleus laminaris, low | 30.3 | 24.8 | 23.9 | 19.9 | −12.7 | 2.3 | *Kuba et al., 2014* |
| Chicks nucleus laminaris, middle | 28.8 | 24.8 | 14.4 | 28.3 | / | 2.8 | *Kuba et al., 2014* |
| Chicks nucleus laminaris, high | 26.5 | 26.6 | 9.8 | 50.1 | −14.3 | 1.8 | *Kuba et al., 2014* |

*Initial = E15, final = P3-7.

structural density of Nav channels did not change. All else being equal, and in the absence of strong AIS hyperpolarization (e.g. Kv7), we then expect that the spike threshold shifts by −5.2 mV (using $k$ = 5 mV; see *Table 2*). The reported change was −4 mV (reported with p<0.05), a fairly good match.

Another study examined the change in AIS geometry through development in the chick nucleus laminaris (*Kuba et al., 2014*). The voltage threshold was found to decrease by −12 to −14 mV between E15 and P3-7, a very large change. AIS geometry also changed significantly in high frequency neurons, with a distal shift (13 to 45 μm, start position) and a shortening (26 to 10 μm). As we have seen, these two changes go in opposite directions. When we combine them, we find a theoretical expectation of 0.4 mV for the threshold shift. Similarly, we find expected shifts of 1–3 mV for low and middle frequency neurons. Could the discrepancy be due to the AIS being hyperpolarized by Kv channels? In that case, somatic threshold should increase when the AIS moves distally, contrary to what is observed. We conclude from this analysis that observed changes in threshold are likely due not to changes in AIS geometry but possibly to changes in patterns of expression of ionic channels. For example, it has been observed in other preparations that Nav1.6 channels appear later than Nav1.2 channels during the course of development (*Boiko et al., 2003*), and the latter activate at lower voltage (*Rush et al., 2005*).

All the other studies that we have examined were in cultured neurons. Chronic depolarization (48 hr) of cultured hippocampal neurons induces a distal shift of the AIS of about 5 μm (*Grubb and Burrone, 2010*). This is a small shift because, as we have seen, what determines the excitability change is the relative change in middle AIS position, which is about 25% here. Together with a slight shortening of the AIS, theory then predicts a modest threshold shift of −1.1 mV, therefore slightly increased excitability. The authors observed on the contrary a decrease in excitability as measured by threshold current density, but it was mostly due to a change in input resistance. Nonetheless, the voltage threshold was also reported to raise by about 4.3 mV; this was not statistically significant because of substantial inter-cellular variations in threshold. If confirmed, such a change could in principle be due to the displacement of hyperpolarizing Kv7 channels. However, to raise the threshold by 5 mV with a 25% distal shift of the AIS means requires that the AIS is initially hyperpolarized by 20 mV relative to the soma. This seems very large. Other possible factors are changes in channel expression, phosphorylation, or possibly axon diameter.

In cultured hippocampal dentate granule cells (*Evans et al., 2015*), 3 hr of depolarization produce a shortening of the AIS, for which we would expect a threshold raised by about 2.4 mV. Instead, the measured threshold is lowered by −1.1 mV in normal condition, and by −1.9 mV in pro-PKA condition (aimed at re-phosphorylating Nav channels). Both changes were not statistically significant. However, when AIS length and threshold are compared within groups (with and without chronic depolarization, with and without pro-PKA treatment), a significant negative correlation is found in each case, consistent with theory. This suggests that the observed decrease in threshold (if genuine) might be due to other factors than AIS geometry.

Finally, in cultured dopaminergic interneurons of the olfactory bulb, one day of chronic depolarization produces a proximal shift and a small elongation of the AIS (*Chand et al., 2015*). As these two factors tend to counteract each other, the theoretical expectation in this case is a small 0.6 mV shift of the threshold. The measured change was −0.4 mV (for the biphasic neurons), which, given the 1.5 mV standard deviation in measurements, appears consistent.

In summary, all studies that we have examined in cultured neurons and in development report changes in AIS geometry that are expected to produce small threshold changes (−1 to 2 mV). In cultured neurons, reported changes are not statistically significant, which could be interpreted as consistent with the expectation. It could also simply signal the possibility that larger data sets are necessary to observe such changes. Possibly, these changes might be more clearly observed in individual cells (rather than by comparing groups). In development, there is a large decrease in threshold that seems likely due to changes in expression patterns of ionic channels. In the one study that we could examine on structural AIS plasticity in adults, the observed threshold change (4–5 mV) was consistent with expectations.

## Discussion

### Summary

A number of recent studies have documented changes in AIS geometry across development (*Galiano et al., 2012*; *Gutzmann et al., 2014*; *Kuba et al., 2014*), with activity (*Grubb et al., 2011*; *Jamann et al., 2018*), or associated with neurological pathologies (*Buffington and Rasband, 2011*). The functional effect of these changes is not addressed by the classical theory of excitability, which has focused on isopotential models of spike initiation (*Koch, 1999*; *Tuckwell, 1988a*). Spatial aspects of neural biophysics have been studied mainly for dendrites, notably by *Rall (2011)*. Here, we have derived an expression for the somatic threshold as a function of geometrical parameters and Nav conductance density, which is well corroborated by simulations of a biophysical model:

$$V_s = \text{constant} - k \log x_{1/2} - k \log L - k \log g - r_a x.I$$

where $k$ is the activation slope of the Nav channels, $x_{1/2}$ is the middle position of the AIS, $L$ is the AIS length, $g$ is Nav conductance density in the AIS, $r_a$ is the axial resistance per unit length, and $I$ is a non-sodium current entering the AIS at position $x$. Excitability changes caused specifically by changes in the AIS are captured by the somatic voltage threshold, rather than rheobase (which depends on other factors) or axonal threshold (which can remain constant when excitability changes due to AIS currents). According to this theoretical analysis, changes in AIS geometry reported in the literature are expected to produce relatively small changes in threshold (a few mV).

The theory explains seemingly contradictory findings from previous simulation studies. Previous theoretical work (*Brette, 2013*) and model simulations (*Baranauskas et al., 2013*; ; *Raghuram et al., 2019*; *Telenczuk et al., 2017*) claimed that the threshold should decrease when the AIS moves away from the soma, because of increased electrical isolation. This claim is valid under two main conditions: 1) that the neuron operates in the resistive coupling regime, meaning that the proximal axon is thin compared to the somatodendritic compartment, 2) that there are no strong subthreshold currents at the AIS. Thus, *Gulledge and Bravo (2016)* found that the neuron can be more excitable when the AIS is away from the soma when the neuron is small (in that case, the number of dendrites was varied). If axon diameter remains large, then the neuron may indeed not be in the resistive coupling regime any more, in which case there is a positive (but weak) relation between AIS distance and excitability. However, we have shown that, both within and across cell types, smaller neurons tend to also have a thinner AIS. This suggests that the physiological regime is generally the resistive coupling regime (with the qualification that the data we analyzed were not exhaustive, and did not include dendritic size). *Lezmy et al. (2017)* have found in a pyramidal cell model that the neuron could also be more excitable when the AIS moves distally, but only when Kv7 channels were strongly expressed at the AIS. Indeed, in this case, Kv7 channels produce a hyperpolarizing current, which raises the somatic threshold by an amount proportional to AIS distance (the last term of the formula).

### Limitations

To obtain this simple theoretical result, we have made a number of more or less drastic approximations detailed in the Results. Despite these approximations, the formula captures the main aspects of the phenomenon seen in a biophysical model, but there are also some discrepancies. In particular, while the relation between threshold and geometrical factors ($x_{\frac{1}{2}}$ and $L$) is well predicted quantitatively, the sensitivity of threshold to Nav channel conductance density tends to be underestimated (but still with a correct order of magnitude).

There are also assumptions that we have made both in the theory and in the biophysical model. A few important assumptions are: spikes are initiated in the initial segment and produce a 'kink' at somatic spike onset; the somatic membrane potential is constant at the time scale of axonal spike initiation; $Na^+$ and $K^+$ currents are temporally separated at spike initiation. These are based on empirical findings (*Brette, 2015*; *Hallermann et al., 2012*), but they could differ in particular models, in which case this theory may not apply.

In our calculations, we have also assumed that the AIS is cylindrical, mainly because analytical calculations are not possible in the general case. In reality, the axon tapers near the soma. The AIS can also sit on an axon-bearing dendrite (*Kole and Brette, 2018*). As explained in the Results (*Role of*

*axon* morphology), one must then consider not metric distances but axial resistances. If the tapering part overlaps with the AIS, then it is not possible to obtain analytical equations anymore; the formula should then be less quantitatively correct. Perhaps the most important point to keep in mind when examining the effect of AIS displacements over a non-uniform axon is that the total Nav conductance varies if Nav channel density is fixed. To avoid this confounding factor, the formula with total conductance $G$ should be used: $V^*_{\mathrm{soma}} \sim -k\log x_{1/2} - k\log G$.

The theory also assumes that the AIS is electrically close to the soma (i.e. relative to the axonal space length). In some neurons, such as dopaminergic neurons (*González-Cabrera et al., 2017*; *Meza et al., 2018*; *Moubarak et al., 2019*), this assumption may not hold and specific theoretical developments might be necessary.

Finally, we have essentially ignored changes in Nav inactivation state. However, these could potentially make an important contribution, in particular for the effect of axonal currents on threshold. As we have seen, a hyperpolarizing current raises the somatic threshold by resistive coupling, by an amount equal to the relative hyperpolarization induced at the AIS. However, this could also deinactivate Nav channels, which would lower the threshold. This effect would tend to cancel the effect due to resistive coupling, and the magnitude can be similar (*Platkiewicz and Brette, 2011*).

## Changes in axon diameter

Axon diameter can change (slightly) with activity (*Chéreau et al., 2017*), although this has not been demonstrated at the AIS. It may also change on longer time scales, particularly during development (*Leterrier et al., 2017*). As explained in the Results, it is possible to take diameter changes into account in our analysis, resulting in the following extended formula:

$$V^*_{\mathrm{soma}} \sim -k\log x_{\frac{1}{2}} - k\log L - k\log g + k\log d$$

where we neglected the modulation by axonal currents.

As we have seen, AIS diameter tends to be larger in larger neurons. The above formula indicates that the spike threshold can be maintained constant across cell types of very different sizes if position and length are constant in units of the axonal space length, that is, if they scale as $\sqrt{d}$. This suggests that smaller neurons should also have a smaller AIS in order to regulate the spike threshold. For example, the AIS measures just 5-10 µm in cerebellar granule cells (*Osorio et al., 2010*) and about 45 µm in layer 5 pyramidal cortical cells (*Hamada et al., 2016*). Their respective diameters are 0.1-0.2 µm and 1-1.5 µm. In chick nucleus laminaris, low-frequency neurons have a large soma and a long AIS, while high-frequency neurons have a small soma and short AIS (*Kuba et al., 2006*; *Kuba et al., 2005*).

Axon diameter can also vary during development and with activity, by the regulation of neurofilaments (*Costa et al., 2018*; *Laser-Azogui et al., 2015*; *Marszalek et al., 1996*). According to the formula, the specific effect of radial growth of the AIS is to raise the threshold (assuming constant surface density of Nav channels). Naturally, it is more difficult to measure changes in AIS diameter than in length, but this potential source of variation must be kept in mind when interpreting changes in AIS geometry.

## Axo-axonic synapses

We have seen that axonal currents can modulate the threshold at the soma. Although we have only discussed Kv7 channels expressed in the AIS, the theory applies also to synaptic currents, in particular those produced at the axo-axonic synapses made by Chandelier cells onto the AIS of pyramidal cortical neurons (*Fairén and Valverde, 1980*; *Somogyi, 1977*). In this case, the current $I$ in the formula must be understood at threshold, that is: $I = g_s.(E_s - V_{\mathrm{threshold}})$, where $g_s$ is the synaptic conductance and $E_s$ is the reversal potential. Thus, even if the synaptic currents are depolarizing at rest (values of $E_s$ = -60 mV have been reported; *Woodruff et al., 2009*), the effect on excitability is still inhibitory, as long as the reversal potential is below axonal threshold. An additional inhibitory effect can be produced if the conductance is strong (specifically, relative to the axial conductance $1/R_a$, see Materials and methods).

In relation to geometry, the effect of an axo-axonic synapse is stronger if the synapse is more distal, up to the AIS position. Beyond the AIS end position, the effect of the synapse does not increase anymore, because it is the voltage gradient between the soma and AIS that modulates excitability.

In hippocampal neurons, *Wefelmeyer et al. (2015)* observed that chronic depolarization made the AIS move distally, but axo-axonic synapses did not move. As a result, a larger proportion of synapses are between the soma and AIS end, where they can modulate excitability. Thus, theory predicts that the total inhibitory effect is stronger (consistent with that study's conclusions).

## Other aspects of electrical function

In this study, we only examined excitability, that is, the ability to trigger an action potential. However, there are others important aspects of electrical function. In many neurons, the action potential is transmitted and regenerated by somatic Nav channels with higher activation voltage. This could be important for synaptic plasticity (propagating the action potential to the dendrites), but also for intrinsic plasticity (since the nucleus is in the soma). For this transmission to be successful, the current transmitted to the soma at spike initiation must be such as to produce the required depolarization. We have shown theoretically that this current depends primarily on the AIS start position $\Delta$ (*Hamada et al., 2016*); as a first (rough) approximation, it is inversely proportional to AIS position (Ohm's law). Thus, AIS position can modulate the transmitted current very strongly, and indeed this strong modulation appears necessary given the three orders of magnitude of variability in input capacitance of various cell types (3500 pF in some motoneurons *Cormery et al., 2005*, about 3 pF in cerebellar granule cells *Cathala et al., 2003*).

In contrast, observed variations in voltage threshold appear rather modest, especially compared to variations in excitability due to changes in input resistance (again 3 orders of magnitude across cell types). An analysis of a large database of electrophysiological recordings reports a standard deviation in voltage threshold of just 6 mV (*Tripathy et al., 2015*), and this includes considerable variations in methodological conditions (liquid junction potential, measurement method, solutions, etc). This is perhaps to be expected, given that the functional voltage range for electrical function is constrained by the properties of ionic channels (it cannot vary by orders of magnitude). Thus, it is conceivable that AIS position and length are adjusted so that voltage threshold remains stable and transmitted current is adapted to the cell's morphology. This is of course speculative, because other aspects of electrical function could also be considered. For example, action potential speed at the AIS depends on local conductance, which is composed of the axial conductance (therefore AIS position) and of the total Nav conductance (therefore AIS length). Thus, structural plasticity of the AIS might be related to various aspects of electrical function, beyond excitability in the classical sense.

# Materials and methods

## Cable theory

### Sealed end

We consider a cylindrical semi-infinite axon, with a current injected at distance $x$ from the sealed end (no current passing through). The input resistance decomposes into $R(x)^{-1} = R_{\text{proximal}}^{-1}(x) + R_{\text{distal}}^{-1}$. $R_{\text{distal}}$ is the resistance of a semi-infinite axon: $R_{\text{distal}} = r_a\lambda$. $R_{\text{proximal}}(x)$ is the resistance of a short segment of axon of size $x$, with a sealed end. This resistance is (*Tuckwell, 1988b*):

$$R_{\text{proximal}}(x) = \frac{r_a\lambda}{\tanh(x/\lambda)}$$

The ratio $R_{\text{proximal}}(x)/R_{\text{distal}}$ is therefore $(\tanh(x/\lambda))^{-1}$. Thus, for a short piece of axon ($x \ll \lambda$), this ratio is $\lambda/x$, a large number. More precisely:

$$R(x) = r_a\lambda \left(1 + \tanh\left(\frac{x}{\lambda}\right)\right)^{-1}$$

and a Taylor expansion gives $R(x) \approx r_a(\lambda - x)$.

### Killed end

We consider a cylindrical semi-infinite axon with a killed end (open membrane) (*Tuckwell, 1988b*). In this case, the resistance of the proximal segment is

$$R_{\text{proximal}}(x) = r_a \lambda \tanh(x/\lambda)$$

When $x \ll \lambda$, we have $R_{\text{proximal}}(x) \approx r_a x$. The distal axon has resistance $R_{\text{distal}} = r_a \lambda$. Therefore, $R_{\text{proximal}}(x)/R_{\text{distal}} \approx x/\lambda$. A Taylor expansion gives $R(x) \approx r_a x(1 - x/\lambda)$. Thus, the input resistance is approximately proportional to distance.

### Large soma
With a large (but not infinitely large) soma, we simply add the resistance of the soma to the proximal resistance, which yields:

$$R(x)^{-1} \approx (r_a x + R_{\text{soma}})^{-1} + (r_a \lambda)^{-1}$$

## Passive neuron models

Passive models presented in *Figures 1* and *2* consisted of a spherical soma (small: 1 μm; large: 100 μm) and a long and thin cylindrical axon (diameter: 1 μm; length: 2 mm). Specific membrane capacitance is $C_m$ = 0.9 μF/cm$^2$; specific membrane resistance is $R_m$ = 15 000 $\Omega$.cm$^2$; leak reversal potential is $E_L$ = -75 mV; intracellular resistivity is $R_i$ = 100 $\Omega$.cm. With these values, the characteristic space length is about $\lambda \approx 600$ μm. Models were simulated with Brian 2 (*Stimberg et al., 2019*) with 100 μs time step and 2 μm spatial resolution.

## Analysis of patch clamp recordings

We analyzed simultaneous patch clamp recordings of soma and axonal blebs in cortical layer 5 pyramidal neurons provided by *Hu and Bean (2018)*; *Figure 2D-F*. We analyzed all recordings where the axon bleb is less than 200 μm away from the soma (a distance substantially smaller than the characteristic length) and is stimulated by a current pulse (n = 15), and selected the smallest current pulse in each recording. To calculate the local and somatic depolarizations at $t$ = 300 μs, we calculate the median potential between 200 and 400 μs and subtract the baseline, defined as the median potential over the 5 ms before the pulse. The resistances are then obtained by dividing by the current amplitude.

## Analysis of neuroanatomical data

In *Figure 3*, we extracted measurements of AIS diameter from electron microscopy studies, taken at the end of the AIS. Given the optical diffraction limit, it is necessary to consider electron rather than optical microscopy measurements, at least for thin axons. Soma diameter was measured with optical or electron microscopy. Four data points correspond to measurements of mean diameters, listed in *Table 3*. Soma and AIS diameters were taken from different studies (except cat olivary cells), but with matched cell type and species.

We also digitized individual measurements in three studies: anterior horn cells (motoneurons) of human spinal cord, 44–75 years old (*Sasaki and Maruyama, 1992*); ventral horn cells (motoneurons) cat spinal cord, P0-P16 (*Conradi and Ronnevi, 1977*); pyramidal, stellate and Martinotti cells of motor and somatosensory cortex of Rhesus monkeys, young adults (*Sloper and Powell, 1979*).

**Table 3.** Mean diameter of soma and AIS in 4 cell types, extracted from electron microscopy studies.

| | | | |
|---|---|---|---|
| Adult cat olivary cells | Soma | 21.7 μm ± 3.7 μm | (*de Zeeuw et al., 1990*) |
| | AIS | 1.1 μm ± 0.3 μm | (*Ruigrok et al., 1990*) |
| Adult rat CA3 pyramidal cells | Soma | 20.9 ± 3.2 μm | (*Buckmaster, 2012*) |
| | AIS | 1.2 μm ± 0.4 μm | (*Kosaka, 1980*) |
| Adult rat Purkinje cells | Soma | 21.9 μm ± 1.9 μm | (*Takacs and Hamori, 1990*) |
| | AIS | 0.7 μm ± 0.2 μm | (*Somogyi and Hámori, 1976*) |
| Adult mouse cerebellar granule cells | Soma | 5.9 μm ± 0.3 μm | (*Delvendahl et al., 2015*) |
| | AIS | 0.2 μm (no s.d.) | (*Palay and Chan-Palay, 2012*) |

## Simplified model

In *Figure 6C-D*, we used a simplified cable model with only non-inactivating Nav channels as a first check of analytical expressions, similar to *Brette (2013)*. A spherical soma of diameter 50 µm is attached to an axonal cylinder of diameter 1 µm and length 300 µm (soma diameter is in fact irrelevant as the soma is voltage-clamped). Specific membrane capacitance is $C_m$ = 0.9 µF/cm$^2$; specific membrane resistance is $R_m$ = 15 000 Ω.cm$^2$; leak reversal potential is $E_L$ = -75 mV; intracellular resistivity is $R_i$ = 100 Ω.cm. Nav channels are placed at a single position on the axon. We used simple single gate activation dynamics with fixed time constant:

$$I_{Na} = Gm(E_{Na} - V)$$

$$\tau_m . \frac{dm}{dt} = m_\infty(V) - m$$

$$m_\infty(V) = \frac{1}{1 + \exp\left((V_{1/2} - V)/k\right)}$$

where $E_{Na}$ = 70 mV, $k$ = 5 mV, $V_{1/2}$ = -35 mV and $\tau_m$ = 53.6 µs (corresponding to 150 µs before temperature correction, see below). Total conductance G is varied between 200 and 600 nS. To give an order of magnitude, this corresponds to a conductance density of about 2000-6000 S/m$^2$ for a 30 µm long AIS. The model is simulated in voltage-clamp and an action potential is detected when half the Nav channels at the AIS are open. We used the Brian 2 simulator (*Stimberg et al., 2019*) with 5 µs time step and 1 µm spatial resolution.

## Biophysical model

The biophysical model has a simple geometry, consisting of a spherical soma (30 µm diameter), a long dendrite (diameter: 6 µm, length: 1000 µm) and a thin unmyelinated axon (diameter: 1 µm, length; 500 µm). When not specified, the AIS extends from 5 µm to 35 µm from the soma. Specific membrane capacitance is $C_m$ = 0.9 µF/cm$^2$; specific membrane resistance is $R_m$ = 15 000 Ω.cm$^2$; leak reversal potential is $E_L$ = -75 mV; intracellular resistivity is $R_i$ = 100 Ω.cm.

The model includes Nav and Kv channels based on Hodgkin-Huxley type equations:

$$I_{Na} = g_{Na}mh(E_{Na} - V)$$

$$I_{Kv1} = g_{Kv1}n^8(E_K - V)$$

$$I_{Kv7} = g_{Kv7}q(E_K - V)$$

where $E_{Na}$ = 70 mV, $E_K$ = −90 mV. Kv7 channels were only included in *Figure 11*. The Na$^+$ current has a single gate, both for simplicity and because it appears to be empirically adequate (*Baranauskas and Martina, 2006*). The K$^+$ current has 8 gates because it matches empirical data for the Kv1 current (*Kole et al., 2007*). This is important to ensure that the current activates with a delay and thereby does not substantially overlap the Na$^+$ current at spike initiation. Each gating variable $x$ is governed by a standard kinetic equation:

$$\frac{dx}{dt} = \alpha_x(1 - x) - \beta_x x$$

where $\alpha_x$ and $\beta_x$ are opening and closing rates, respectively. For the voltage-dependent rates, we chose expressions that feature a minimal number of parameters with simple interpretation:

$$\alpha_x(V) = \frac{1}{2k_x\tau_x^*} . \frac{V - V_x^{1/2}}{1 - e^{-\left(V - V_x^{1/2}\right)/k_x}}$$

$$\beta_x(V) = -\frac{1}{2k_x\tau_x^*} . \frac{V - V_x^{1/2}}{1 - e^{\left(V - V_x^{1/2}\right)/k_x}}$$

where $x$ is $m$ or $n$. For inactivation ($h$), opening and closing rates are simply exchanged. These expressions have the following properties: the equilibrium value ($x_\infty(V)$) is a Boltzmann function with half-activation value $V_x^{1/2}$ and slope factor $k_x$; the voltage-dependent time constant is a bell curve peaking at $V_x^{1/2}$, where its value is $\tau_x^*$.

Parameter values vary substantially between studies. For Nav channels of proximal axons, empirically measured slope factors vary between 5 and 8 mV for activation and 5-9 mV for inactivation (*Engel and Jonas, 2005*; *Hu et al., 2009*; *Kole and Stuart, 2008*; *Schmidt-Hieber and Bischofberger, 2010*). When fitted on the hyperpolarized part important for spike initiation, the activation curve of Nav channels tend to be lower, as explained in *Platkiewicz and Brette (2010)*; *Figure 9* and observed empirically (*Baranauskas and Martina, 2006*; *Figure 2*). Therefore, we simply used rounded values, $k_m = k_h = 5$ mV. For half-activation values, we also used rounded values consistent with the literature, with voltage-dependent curves hyperpolarized by 5 mV in the AIS compared to the soma (see *Table 1*). We used $\tau_m^* = 150$ μs as in *Schmidt-Hieber and Bischofberger (2010)*, noting that such short time constants are challenging to measure experimentally, especially in axons. For inactivation, we used $\tau_h^* = 5$ ms. This value was chosen so that the Na$^+$ current during the action potential shows a small overlap with the K$^+$ current, as experimentally observed (*Hallermann et al., 2012*; *Figure 4E, F*). Conductance densities were set as stated in *Table 1*, following experimental and modeling studies (*Hallermann et al., 2012*; *Hu et al., 2009*; *Kole and Stuart, 2008*; *Lorincz and Nusser, 2010*). Finally, rates were corrected for temperature. Recordings on which parameter values are based were done at T=23°C, and we set the simulation temperature at 33°C by applying a temperature factor $Q_{10}^{(33-23)/10}$, with $Q_{10} = 2.8$ (*Baranauskas and Martina, 2006*).

Densities of Kv1 channels were set similarly to previous studies (*Hallermann et al., 2012*; *Hu et al., 2009*; *Kole et al., 2008*) (see *Table 1*). Kv1 channels activate quickly and inactivate slowly (*Kole et al., 2007*). We did not include inactivation because it has no influence on spike initiation. Parameter values were obtained by least square fitting a Boltzmann function to the activation curve ($n_\infty^8(V)$) obtained from recordings of axonal outside-out blebs of layer 5 pyramidal neurons (*Kole et al., 2007*). As data were recorded at 33°C, transition rates were not corrected for temperature.

In *Figure 11*, we added a Kv7 conductance on the distal half of the AIS. Kv7 channels produce the M-current, which activates slowly at low voltage and does not inactivate (*Battefeld et al., 2014*). We used parameter values rounded from *Battefeld et al. (2014)*.

## Spike threshold with a point AIS

A formula for the spike threshold with a point AIS has been derived in *Brette (2013)*. Here, we put a simpler alternative derivation based on a rescaling argument.

In the hyperpolarized range, the Na$^+$ current changes approximately exponentially with voltage (*Baranauskas and Martina, 2006*; *Hodgkin and Huxley, 1952b*): $I_{Na} \approx G \exp(V/k)(E_{Na} - V_{1/2})$, where $V$ is the axonal membrane potential, $G$ is the total available Na$^+$ conductance, $k$ is the Boltzmann slope factor of Nav channels and $V_{1/2}$ is the half-activation voltage of Nav channels. As the soma is a current sink (hypothesis of resistive coupling theory), this current flows towards the soma as a resistive current: $I_{\text{axial}} = (V - V_s)/R_a$, where V$_s$ is the somatic membrane potential and $R_a$ is the axial resistance between soma and AIS. Therefore, we have the following identity:

$$R_a G(E_{Na} - V_{1/2}) \exp(V/k) = V - V_s$$

With the following change of variables:

$$U = V + k \log(R_a G(E_{Na} - V_{1/2}))$$

$$U_s = V_s + k \log(R_a G(E_{Na} - V_{1/2}))$$

we obtain:

$$\exp(U/k) = U - U_s$$

This equation is now independent of $G$ and $R_a$. We denote $U^*$ the threshold for this equation, that is, such that the equation has a bifurcation when $U_s = U^*$. For the original equation, this corresponds to a bifurcation when $V_s = U^* - k \log \left( R_a G \left( E_{Na} - V_{1/2} \right) \right)$.

## Spike threshold with an extended AIS starting from the soma

We consider a cylindrical axon of diameter $d$. The AIS has length L and starts from the somatic end. It has a uniform density of Nav channels. The total Nav conductance is

$$G = g \times \pi d L$$

where $g$ is the surface conductance density. We neglect leak and $K^+$ currents, as well as all time-varying phenomena. The cable equation then becomes:

$$\frac{d^2 V}{dx^2} \propto -g \mathrm{e}^{V/\mathrm{k}}$$

with boundary conditions $V(0) = V_s$ (somatic potential) and $V'(L) = 0$ (no axial current flowing towards the distal axon). In units of the AIS length $L$, this equation reads:

$$\frac{d^2 V}{d(x/L)^2} \propto -g L^2 \mathrm{e}^{\frac{V}{k}} = -\exp \left( \frac{V + k \log g L^2}{k} \right)$$

By the same argument as in the previous section, it follows that the threshold varies with $g$ and $L$ as

$$V_s = \mathrm{constant} - k \log g - 2k \log L$$

This is equivalent to:

$$V_s = \mathrm{constant} - k \log G - k \log L$$

where $G$ is the total Nav conductance ($G = \pi d L$ for a cylindrical axon), for a different constant.

We now calculate the constant. With the proportionality factor, the equation is approximately:

$$\frac{d^2 V}{d(x/L)^2} = -\pi d r_a g L^2 \left( E_{Na} - V_{1/2} \right) \mathrm{e}^{(V - V_{1/2})/\mathrm{k}}$$

where $r_a$ is resistance per unit length and $V_{1/2}$ is the half-activation voltage of Nav channels. Here the driving force $(E_{Na} - V)$ has been approximated by $\left( E_{Na} - V_{1/2} \right)$ as in **Brette (2013)**. We now write the following change of variables:

$$U = \left( V - V_{1/2} \right)/k + \log \left( \pi d r_a g L^2 \left( E_{Na} - V_{\frac{1}{2}} \right)/k \right)$$

$$y = x/L$$

and we note $U' = dU/dy$. That is, voltage is in units of $k$ and space is in units of AIS length $L$. The rescaled cable equation is:

$$U'' + e^U = 0$$

with the boundary conditions:

$$U(0) = U_0 = \left( V_s - V_{1/2} \right)/k + \log \left( \pi d r_a g L^2 \left( E_{Na} - V_{\frac{1}{2}} \right)/k \right)$$

$$U'(1) = 0$$

This equation is analytically solvable, with general solution

$$U(y) = \log\frac{c_1}{2} - 2\log\left(\cosh\left(\frac{1}{2}\sqrt{c_1(c_2+y)^2}\right)\right)$$

From $U'(1) = 0$, it follows that $c_2 = -1$.
We then obtain for the boundary condition at 0:

$$U(0) = U_0 = \log\frac{c_1}{2} - 2\log\left(\cosh\left(\frac{\sqrt{c_1}}{2}\right)\right)$$

which defines $c_1$ as an implicit function of $U_0$. We look for a bifurcation, that is, a value of $U_0$ when the number of solutions changes. This is obtained by setting the derivative of the right hand-side to 0, which gives:

$$\frac{\sqrt{c_1}}{2}\tanh\left(\frac{\sqrt{c_1}}{2}\right) = 1$$

The solution can be calculated: $\sqrt{c_1}/2 \approx 1.2$, giving $c_1 \approx 5.8$. Finally, substituting this value in the above equation gives $U_0 \approx -0.13$. This is the spike threshold for the rescaled cable equation. Back to the original dimensions, we obtain:

$$V_s = V_{1/2} - 0.13k - k\log\left(r_a\left(E_{Na} - V_{\frac{1}{2}}\right)/k\right) - k\log(\pi dg) - 2k\log L$$

This is the same equation as for a point AIS with the same total conductance $G$ at position $L$, except that the term *-0.13 k* replaces *-k*. Thus, the difference in threshold between an extended AIS of length $L$ starting from the soma and a point AIS at position $L$ is *0.87 k* (4.3 mV if $k$ = 5 mV). Therefore, the threshold of the extended AIS is the same as a point AIS placed at position $x = e^{-0.87}L \approx 0.42\,L$, which is near the middle of the extended AIS. The error made by placing the equivalent point AIS at the middle point $x = L/2$ is $k\log(0.5/0.42) \approx 0.9$ mV (with $k$ = 5 mV).

The expression of $U(y)$ allows us to calculate the potential along the axon at threshold, and in particular at the AIS end, where the expression simplifies: $U(1) = \log(c_1/2) \approx 1.06$. We can see that the threshold at the AIS end is above the somatic threshold by about *1.2 k* (1.06+0.13). This is consistent with simultaneous patch clamp measurements at the soma and AIS (*Kole et al., 2008*).

## Spike threshold with an extended AIS starting away from the soma

We apply the same strategy for the more general case where the AIS starts at a distance $\Delta$ away from the soma. We choose the origin of $x$ at the AIS start, so that we obtain exactly the same cable equation as before, except the boundary condition at $x = 0$ now expresses the fact that the piece of axon between the soma and AIS is purely resistive. This implies that the potential varies linearly with distance, and therefore:

$$V(0) = V_s + \Delta\frac{dV}{dx}(0)$$

Thus, we obtain the same solution as previously except for the boundary condition at 0:

$$U(0) = U_0 = \log\frac{c_1}{2} - 2\log\left(\cosh\left(\frac{\sqrt{c_1}}{2}\right)\right) - \frac{\Delta}{L}\sqrt{c_1}\tanh\frac{\sqrt{c_1}}{2}$$

As before, to find the bifurcation point we set the derivative of the right hand-side (with respect to $c_1$) to 0, and obtain:

$$f(z) \equiv \left(1 + \frac{\Delta}{L}\right)z\tanh z + \frac{\Delta}{L}z^2\left(1 - \tanh^2 z\right) - 1 = 0$$

where $z = \sqrt{c_1}/2$. This defines $z$, $c_1$ and therefore $U_0$ as implicit functions of $\Delta/L$, which can be calculated numerically (which we did in *Figure 9*). The somatic threshold is then:

$$V_s = V_{1/2} + kU_0(\Delta/L) - k\log\left(r_a\left(E_{Na} - V_{\frac{1}{2}}\right)/k\right) - k\log(\pi dg) - 2k\log L$$

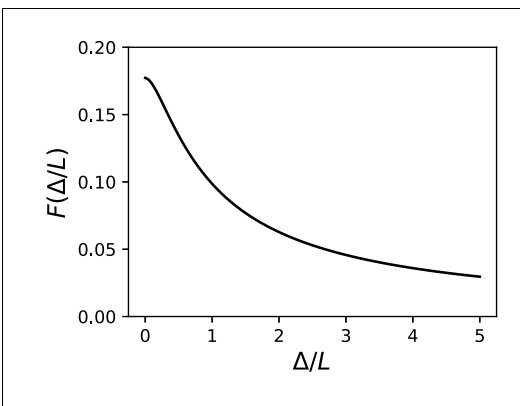

**Figure 13.** Corrective term $F(\Delta/L)$. The threshold of an extended AIS differs from that of point AIS with the same total conductance placed at the midpoint by at most $kF(\Delta/L)$.

The somatic threshold for a point AIS with the same total conductance placed at the midpoint $x^* = \Delta + L/2$ is:

$$V_s^* = V_{1/2} - k - k\log\left(r_a\left(E_{Na} - V_{\frac{1}{2}}\right)/k\right) - k\log(\pi dgL) - k\log x^*$$

The difference is:

$$V_s - V_s^* = k\left(U_0\left(\frac{\Delta}{L}\right) + 1 + \log\left(\frac{\Delta}{L} + \frac{1}{2}\right)\right) \equiv kF\left(\frac{\Delta}{L}\right)$$

The variable $\Delta/L$ varies between 0, where the AIS starts from the soma, and $+\infty$, where the AIS is a single point. *Figure 13* shows that the function F is a monotonously decreasing function of $\Delta/L$. When $\Delta/L = 0$, the AIS starts from the soma, and we have seen in the previous section, $V_s - V_s^* \approx 0.9$ mV (0.17 $k$). When $\Delta/L = +\infty$, the AIS is a single point and therefore $V_s = V_s^*$. Thus, the somatic threshold of the extended AIS is approximately equivalent to the threshold of a point AIS with the same total conductance, placed at the midpoint $x_{1/2}$, with a precision of about $0.17\,k \approx 0.9$ mV.

## Effect of an axonal current on spike threshold

The effect of an axonal current on the spike threshold of a point AIS has been derived in *Brette (2013)* (Supplementary Text). We show that this extends to an extended AIS, where a current $I$ is injected at the start of the AIS. In that case, the cable equation is unchanged, but the boundary condition at the AIS start ($x = 0$) now includes the current:

$$V(0) = V_s + \Delta\frac{dV}{dx}(0) + R_a I$$

where $R_a$ is the axial resistance between soma and AIS start. Thus, inserting this current is equivalent to shifting the somatic potential by an amount $I/R_a$. Thus, the bifurcation occurs when $V_s + R_a I = V_s^*$, where $V_s^*$ is the somatic threshold without modulation ($I = 0$). The somatic threshold with modulation is therefore $V_s^* - R_a I$. At threshold, the boundary condition is independent of $I$, and therefore the axonal voltage at threshold does not depend on $I$.

If the current is injected at the AIS end, then the boundary condition at the AIS end becomes:

$$\frac{dV}{dx}(L) = r_a I$$

The cable equation can still be solved analytically as before. However, it does not lead to any simple expression of threshold as a function of $I$. It is found numerically that the somatic threshold changes almost (but not exactly) linearly with $I$, and the threshold at the AIS end varies slightly with $I$, in the other direction (decreases for a strong hyperpolarizing current).

If current is uniformly injected over the AIS, then boundary conditions are unchanged but the current density is inserted in the cable equation. To our knowledge, it has no analytical solution.

The theoretical analysis above applies to an injected current. The effect of inserting a conductance, that is, $I = g^*(E - V)$, can be understood in the point AIS model by noting that the conductance $g^*$ is in parallel with the axial resistance $R_a$. Therefore, it is equivalent to replacing $R_a$ by $\left(R_a^{-1} + g^*\right)^{-1}$. As long as $g^*$ is small compared to $1/R_a$, this effect is negligible. That is, the current-based theory holds, with $I = g^*(E - V_{\text{threshold}})$, where $V_{\text{threshold}}$ is the AIS threshold. If $g^*$ is large, the effective change in $R_a$ must be taken into account.

## Data availability

Code generating all figures is available at: https://github.com/romainbrette/AIS-geometry-and-excitability-2019 (*Goethals and Brette, 2020*; copy archived at https://github.com/elifesciences-publications/AIS-geometry-and-excitability-2019).

## Acknowledgements

We thank Marcel Stimberg for assistance with simulations, Bruce Bean and Wenqin Hu for providing their electrophysiological data, and Christophe Leterrier and Boris Barbour for discussions.

## Additional information

### Funding

| Funder | Grant reference number | Author |
|---|---|---|
| Agence Nationale de la Recherche | ANR-14-CE13-0003 | Sarah Goethals Romain Brette |
| Ecole des Neurosciences de Paris | | Sarah Goethals Romain Brette |

The funders had no role in study design, data collection and interpretation, or the decision to submit the work for publication.

### Author contributions

Sarah Goethals, Conceptualization, Data curation, Software, Formal analysis, Investigation, Visualization, Methodology; Romain Brette, Conceptualization, Software, Formal analysis, Supervision, Funding acquisition, Validation, Investigation, Methodology

### Author ORCIDs

Romain Brette (iD) https://orcid.org/0000-0003-0110-1623

### Decision letter and Author response

Decision letter https://doi.org/10.7554/eLife.53432.sa1
Author response https://doi.org/10.7554/eLife.53432.sa2

## Additional files

### Supplementary files

• Transparent reporting form

### Data availability

Code to generate all figures is available on GitHub: https://github.com/romainbrette/AIS-geometry-and-excitability-2019 (copy archived at https://github.com/elifesciences-publications/AIS-geometry-and-excitability-2019). Electrophysiological data analyzed in Fig. 2 has been uploaded on Zenodo (http://doi.org/10.5281/zenodo.3539296), on behalf of Prof. Bean (Data from Hu & Bean, 2018). Digitized data used in Fig. 3 have been uploaded on GitHub (link above).

The following previously published dataset was used:

| Author(s) | Year | Dataset title | Dataset URL | Database and Identifier |
|---|---|---|---|---|
| Hu W, Bean B | 2019 | Responses to axonal current injection in cortical layer 5 pyramidal neurons | http://doi.org/10.5281/zenodo.3539296 | Zenodo, 10.5281/zenodo.3539296 |

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
