## [Decision Letter]

**Acceptance summary:**

This is an interesting theoretical study that examines axon initial segment (AIS) properties on neuron excitability. From a thorough mathematical analysis, the authors are able to explain seemingly contradictory results of previous computational studies.

**Decision letter after peer review:**

Thank you for submitting your article "Theoretical relation between axon initial segment geometry and excitability" for consideration by *eLife*. Your article has been reviewed by three peer reviewers, one of whom is a member of our Board of Reviewing Editors, and the evaluation has been overseen by Ronald Calabrese as the Senior Editor. The following individual involved in review of your submission has agreed to reveal their identity: Farzan Nadim (Reviewer #2).

The reviewers have discussed the reviews with one another and the Reviewing Editor has drafted this decision to help you prepare a revised submission.

This is a solid theoretical/computational study that examines axon initial segment (AIS) properties on neuron excitability. Specifically, on how the position of the AIS in vertebrate neurons influences the excitability of neurons from the viewpoint of the soma, the site of synaptic integration. The authors provide a thorough mathematical analysis that explains the seemingly contradictory results of previous computational studies. The analysis shows that a distal shift of the AIS increases excitability, except in the presence of hyperpolarizing currents (low-threshold potassium currents or inhibitory synaptic input) in the distal AIS, in which case the shift may decrease excitability. This is an excellent, systematic study of a complicated, timely and important issue.

However, two major concerns were raised and need to be addressed. It was also noted that the code for the figures is claimed to be on Github, but the link is currently broken.

Essential revisions:

1) The significance of this theoretically solid study does not come across from the Introduction or Discussion section.

The Introduction starts with facts about AIS in vertebrate axons with some reference to disease. It does describe the question properly and says what the authors plan to do in this study. What the Introduction fails to do, is to provide the reader with the significance of the question, a proper scientific (not disease) context. That is, why should I as a scientist care about the location of the AIS? Seems like some esoteric thing that theoreticians worry about and is not of any consequence to neuroscience at large and certainly not of importance for the general scientific readership of *eLife*. The potential significance is realized after reading the Results section. I suggest that the authors start with a more general statement of significance which is less jam-packed with citations and introduces the readers to the importance of the question. I also suggest that they come back to this point of general significance in the Discussion section.

2) There is relative lack of attention paid to the role of the axon morphology distal to the AIS. As the authors note in subsection “Small soma, or sealed end condition”, the space constant in pyramidal neurons is around 500um, but since it varies proportional to the square root of the diameter (subsection “Changes in axon diameter”), along with other factors, it may be expected to range up to millimetres in some neurons. This means that it could constitute a large current sink, perhaps competing with the soma. A second issue is that axon morphology can show complicated branching patterns within the electrotonic range from the AIS (Wang, 2019), and since the branching cable structure affects the set of voltage time constants (Rall, 1969), the short-term response to injected current inputs in a simple cable, especially if truncated, may not match the voltage responses in real neurons (or models with longer, branching axons). In the biophysical model in this study the authors use a cylindrical axon of length 500um (subsection “Biophysical model”), which is far from infinite length relative the space constant, and so may underestimate current flow and dynamics in the distal direction. What are the authors views on this issue?

In essence, the key issue is whether having an extended axon affects the qualitative scaling laws the authors describe for somatic voltage threshold. I just don't have an intuition for what to expect: on the one hand it might only have a quantitative effect, shifting the numbers a bit. But on the other hand it might move the neuron model into the "small soma" or "sealed-end condition" in the authors' phrasing, which would dramatically change the paper's conclusions. The authors should estimate the likelihood of this outcome in a simulation – it could be done by extending one of their presented model (rather than making a new model from scratch).

---

## [Author Response]

Essential revisions:1) The significance of this theoretically solid study does not come across from the Introduction or Discussion section.The Introduction starts with facts about AIS in vertebrate axons with some reference to disease. It does describe the question properly and says what the authors plan to do in this study. What the Introduction fails to do, is to provide the reader with the significance of the question, a proper scientific (not disease) context. That is, why should I as a scientist care about the location of the AIS? Seems like some esoteric thing that theoreticians worry about and is not of any consequence to neuroscience at large and certainly not of importance for the general scientific readership of eLife. The potential significance is realized after reading the Results section. I suggest that the authors start with a more general statement of significance which is less jam-packed with citations and introduces the readers to the importance of the question. I also suggest that they come back to this point of general significance in the Discussion section.

We have rewritten the first paragraph of the Introduction as suggested. A number of citations have been removed (in particular, for activity-dependent plasticity, we kept only the reviews). We also added a short text at the beginning of the Discussion section.

2) There is relative lack of attention paid to the role of the axon morphology distal to the AIS. As the authors note in subsection “Small soma, or sealed end condition”, the space constant in pyramidal neurons is around 500um, but since it varies proportional to the square root of the diameter (subsection “Changes in axon diameter”), along with other factors, it may be expected to range up to millimetres in some neurons. This means that it could constitute a large current sink, perhaps competing with the soma. A second issue is that axon morphology can show complicated branching patterns within the electrotonic range from the AIS (Wang, 2019), and since the branching cable structure affects the set of voltage time constants (Rall, 1969), the short-term response to injected current inputs in a simple cable, especially if truncated, may not match the voltage responses in real neurons (or models with longer, branching axons). In the biophysical model in this study the authors use a cylindrical axon of length 500um (subsection “Biophysical model”), which is far from infinite length relative the space constant, and so may underestimate current flow and dynamics in the distal direction. What are the authors views on this issue?In essence, the key issue is whether having an extended axon affects the qualitative scaling laws the authors describe for somatic voltage threshold. I just don't have an intuition for what to expect: on the one hand it might only have a quantitative effect, shifting the numbers a bit. But on the other hand it might move the neuron model into the "small soma" or "sealed-end condition" in the authors' phrasing, which would dramatically change the paper's conclusions. The authors should estimate the likelihood of this outcome in a simulation – it could be done by extending one of their presented model (rather than making a new model from scratch).

To address these remarks, we have added a new section and a new figure (Figure 13) at the end of the Results section on the role of axon morphology. In this process, some of the text previously in the Discussion section on axon diameter was moved up.

Regarding the first point about axon diameter, there are two aspects. (1) The size of the soma seems to scale with AIS size such that electrical properties are conserved. (2) The proportion of current flowing towards the soma vs. the distal axon depends on the relative resistance in these two directions. This ratio is conserved if the position is constant in units of the axon space constant. Therefore, if AIS geometry scales with axon space constant, then the electrical configuration should be conserved.

We have added a short theoretical analysis of the role of the distal axon. We find in particular that extending the axon should have no impact, but myelinating it slightly lowers the threshold, in a way that can be estimated. The idea is that the leak towards the distal end is formally the same as a conductance applied at the distal end of the AIS, and therefore we can apply the results already obtained.

We have also analyzed axonal branching, when branching occurs after the AIS and before the AIS (with an axon-carrying dendrite). Overall, the quantitative changes compared to the simple unbranched axon are small.